# Graphene memristive synapses for high precision neuromorphic computing

Thomas F. Schranghamer[1], Aaryan Oberoi [1] & Saptarshi Das [1,2,3 ✉]

Memristive crossbar architectures are evolving as powerful in-memory computing engines for artificial neural networks. However, the limited number of non-volatile conductance states offered by state-of-the-art memristors is a concern for their hardware implementation since trained weights must be rounded to the nearest conductance states, introducing error which can significantly limit inference accuracy. Moreover, the incapability of precise weight updates can lead to convergence problems and slowdown of on-chip training. In this article, we circumvent these challenges by introducing graphene-based multi-level (>16) and non-volatile memristive synapses with arbitrarily programmable conductance states. We also show desirable retention and programming endurance. Finally, we demonstrate that graphene memristors enable weight assignment based on $k$-means clustering, which offers greater computing accuracy when compared with uniform weight quantization for vector matrix multiplication, an essential component for any artificial neural network.

[1] Department of Engineering Science and Mechanics, Pennsylvania State University, University Park, PA 16802, USA. [2] Department of Materials Science and Engineering, Pennsylvania State University, University Park, PA 16802, USA. [3] Materials Research Institute, Pennsylvania State University, University Park, PA 16802, USA. ✉email: sud70@psu.edu

The recent decline in complementary metal-oxide-semiconductor (CMOS) technology after almost five decades of relentless growth necessitates alternate computing methods to circumvent existing challenges[1]. A subject of great interest in this regard is the human brain. While powerful supercomputers can rival or even exceed the brain in number of operations performed per second, the brain is indisputably superior in terms of energy and area efficiency, capable of performing anywhere from 5 trillion to 5 quadrillion operations per Watt and only taking up $0.0012 \, m^3$ in volume[2]. In comparison, IBM's supercomputer Summit can only perform approximately 10 billion operations per Watt while taking up an area of over $850 \, m^2$[3]. Artificial neural networks (ANNs) seek to emulate the efficiency of the brain by directly mimicking its most fundamental unit: neuron-to-neuron connections via synapses. However, even the most sophisticated chips based on ANNs, such as IBM's TrueNorth[4], lack the ability to be scaled up to the full capacity of a human brain without becoming inordinately power hungry and area-inefficient[5]. The traditional von Neumann architecture that operates on the basis of physical separation between logic and memory is inherently incapable of scaling ANNs with millions of synaptic weights.

The motivation behind biologically-inspired computing architecture lies in the ability of such systems to continuously adapt to external stimuli that varies with time[6]. For ANNs, such learning is obtained by modulating the synaptic weights assigned to the connections between neurons, allowing for the overall connectivity of the network to be reconfigured[7]. To properly reproduce this functionality from biological neurons, ANNs require a device capable of changing and retaining its synaptic weight (resistance/conductance) upon experiencing synaptic activity (the application of a current or bias) while also demonstrating analog behavior (possessing several resistance/conductance states). In this context, modern ANNs have progressed tremendously when compared to the first computational model developed by McCulloch and Pitts[8,9], with different ANNs being classified according to their respective network architectures and connectivity structures. ANNs possess a large number of computational layers, and those possessing greater than three layers are often referred to as deep neural networks (DNNs). The layers of greatest interest for the purposes of this paper are fully connected (FC) layers that appears in all forms of ANNs. These are layers wherein all outputs from a single layer are connected to all inputs of the next layer, allowing for this next layer to compute the weighted sum of all the outputs. This is typically done by performing vector-matrix multiplication (VMM) upon the outputs[8]. This process is extremely energy inefficient using CMOS technology in conjunction with the traditional von Neumann computing architecture. Recent research has shown that higher efficiency can be achieved by exploiting a crossbar array architecture and utilizing a direct weight update scheme based on physical laws. Each crosspoint in the array is composed of a material with adjustable conductance, $G$, essentially making each crosspoint an analog non-volatile memory cell. By mapping the weight matrices of FC layers to the conductance matrices of the crossbar arrays, VMM can be performed at lower latency and thus avoid the von Neumann bottleneck, i.e., data shuttling between memory and compute. The development of such devices is aided by resistive random access memory (RRAM), or memristors, that display a programmable conductance capable of being changed via the application of short (<1 s), high amplitude (>1 V) voltage pulses[10–13]. Most memristors are, however, binary since they possess only two resistance states: a high resistance state (HRS), in which the device is considered to be off, and a low resistance state (LRS), in which the device is considered to be on. Analog operation, mentioned previously as possessing several resistance states, is far more preferred due to its enhanced accuracy (through minimization of quantization error) over binary operation, however the difficulty of operating memristors in an analog fashion is a significant limitation that may hinder its hardware implementation. One solution is to implement analog operations using binary devices. But this naturally leads to high computational and memory costs, limiting the application of ANNs in situations with limited storage and computing power, a prime example being portable devices[14,15]. In order to increase power, area, and computational efficiency, weights are often quantized into lower bits. By performing mathematical operations at lower-precisions (i.e., 8-bit integer operations as opposed to 32-bit floating point operations), ANNs consume less energy and increase efficiency while also requiring less memory storage. A known downside of this approach is a loss of accuracy due to non-idealities (namely quantization errors and noise) generated by the weight quantization process, which can negatively impact an ANN's ability to converge[16,17].

In this article, we experimentally demonstrate a non-volatile graphene-based resistive memory device which is capable of achieving in excess of 16 conductance states. While non-volatile graphene memory is not a new concept[18–23], most explorations into graphene memory are unable to realize more than 2 memory states (1-bit) on a single device. We also show that the graphene memristive synapses possess desirable retention and switching endurance and also allow for the hardware implementation of quantization through $k$-means clustering, resulting in enhanced accuracy when compared to the uniform weight quantization used by other synaptic devices. Overall, our demonstration of multi-bit and non-volatile graphene memristive synapses can be transformative for the realization of area and energy efficient hardware for neuromorphic computing and for the integration of ANNs with emerging technologies such as the Internet of Things (IoT)[24].

## Results

**Non-volatile and multi-bit graphene-based memristors**. We have achieved programmable conductance in graphene field effect transistor (GFET) devices similar to that seen in oxide-based memristors. To fabricate the GFETs, large-area chemical vapor deposition (CVD) grown graphene was transferred onto a 50 nm alumina ($Al_2O_3$) substrate, which acts as a back-gate oxide, on a stack of Pt/TiN/p$^{++}$-Si, which functions as a back-gate electrode. The use of 50 nm $Al_2O_3$ as the back-gate oxide, when compared to conventional 300 nm $SiO_2$, was motivated by the high relative dielectric constant (~10) of $Al_2O_3$ that allows for better electrostatic control of the GFET. Each GFET used for the experiments was fabricated with a channel length (L) and channel width (W) of 1 μm and 0.5 μm, respectively. Further fabrication details, including the specifics of the transfer process used, can be found in the "Methods" section. Figure 1a, b, respectively, show the schematic and scanning electron microscope image of a representative GFET. Figure 1c shows the Raman spectrum of the graphene channel, taken at a wavelength of 532 nm. The peak at approximately $1600 \, cm^{-1}$ is known as the G-band and is found in all sp$^2$ carbon materials as a result of C–C bond stretching. The existence of a strong peak at a Raman shift value of $2500–2800 \, cm^{-1}$ indicates the presence of single-layer graphene, with the peak itself being referred to as the 2D-band. Notably, the Raman spectrum shown here lacks a peak at approximately $1400 \, cm^{-1}$, as well as a sub-peak directly adjacent to the G-band. These peaks, known as the D-band and D'-band, respectively, are indicative of disorder/impurities in the sp$^2$ structure of graphene. The absence of these peaks thus indicates that the graphene used in the GFETs discussed in this paper is of high quality in addition to being

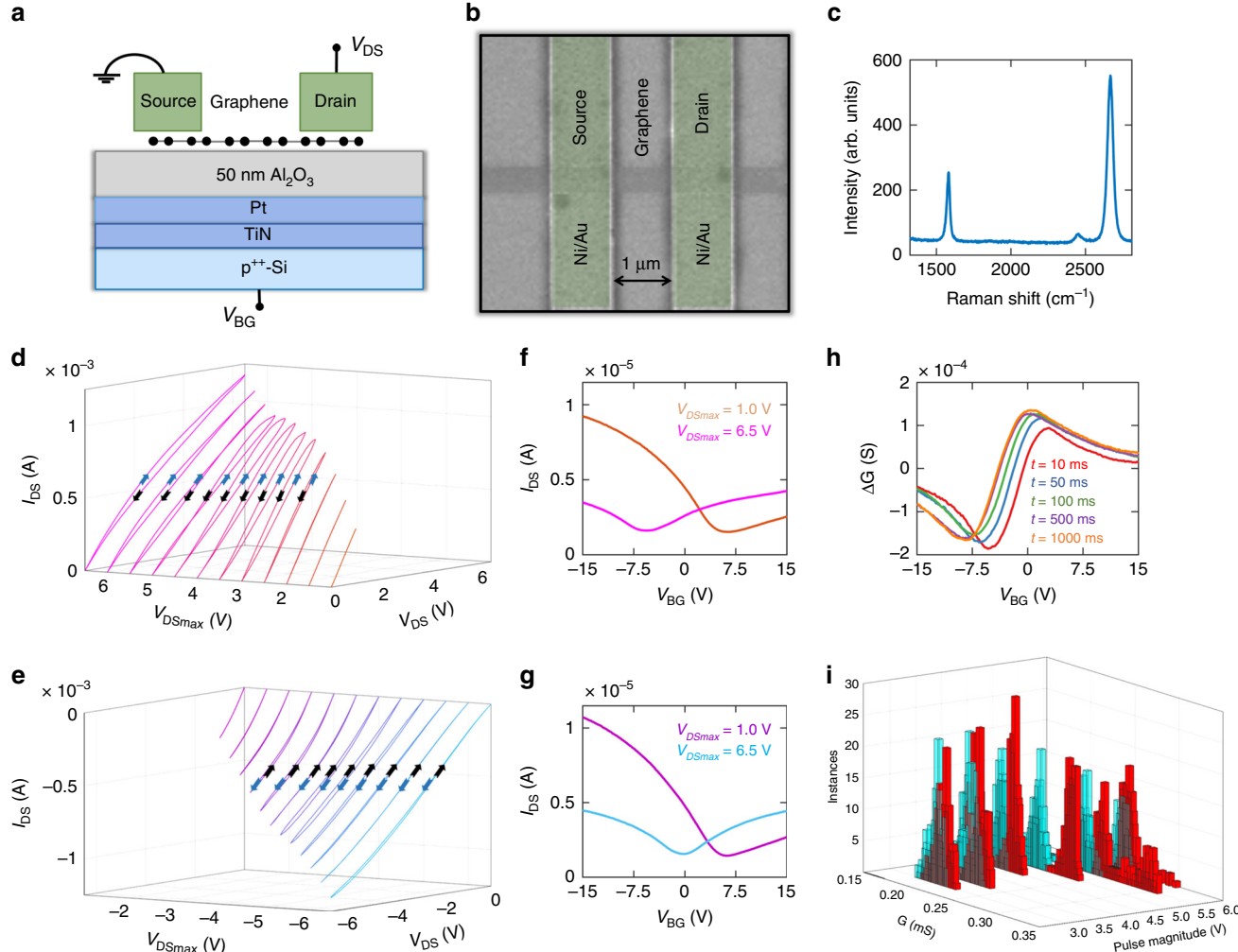

**Fig. 1 SET process for graphene memristors. a** Schematic and (**b**) false-colored scanning electron microscope (SEM) image of a graphene memristor. **c** Raman spectrum of the graphene channel, taken at a wavelength of 532 nm. Output characteristics (i.e., the source-to-drain current ($I_{DS}$) versus the drain-to-source voltage ($V_{DS}$)) of an as-fabricated graphene field effect transistor (GFET) at $V_{BG} = 0$ V for different $V_{DS}$ sweep ranges (denoted by $V_{DSmax}$), from (**d**) 1 V to 6.5 V and (**e**) −1 V to −6.5 V in steps of 0.5 V. The arrows denote the sweep direction (blue for the forward sweep and black for the reverse sweep). In either case, the hysteresis window initially increases with increasing $V_{DSmax}$ before reversing direction and starting to decrease past $V_{DSmax} = 5$ V for (**d**) and $V_{DSmax} = -5.5$ V for (**e**). These results indicate switching between states of lower and higher conductance in GFETs. Transfer characteristics at $V_{DS} = 10$ mV following the sweeps of (**f**) $V_{DSmax} = 1$ V and $V_{DSmax} = 6.5$ V and (**g**) $V_{DSmax} = -1$ V and $V_{DSmax} = -6.5$ V. Sweeping the GFET to a higher positive $V_{DSmax}$ results in a large shift of the Dirac voltage from $V_{Dirac} = 6.4$ V to $V_{Dirac} = -5.8$ V, making the GFET more n-type, whereas sweeping the GFET to a higher negative $V_{DSmax}$ results in a smaller shift from $V_{Dirac} = 6.5$ V to $V_{Dirac} = -0.2$ V, making the GFET more ambipolar. **h** Difference between the conductance of two states as a function of $V_{BG}$ after the sequential application of positive and negative $V_{DS}$ pulses of magnitude 5 V for different pulse durations ($t$) at $V_{DS} = 10$ mV. **i** Switching endurance. Histogram of conductance distributions following 200 cycles of SET (red) and RESET (blue) pulses of different magnitudes. Conductance states obtained using $V_{DS}$ pulses of magnitude 5 V display both a relatively large difference in conductance and a switching endurance >200 cycles. These experiments demonstrate the ability to SET and RESET conductance states in graphene by applying $V_{DS}$ pulses of opposite polarity, making it attractive for non-volatile memory (NVM) applications.

single-layer. Figure 1d, e display the output characteristics, i.e., the source-to-drain current ($I_{DS}$) versus the drain-to-source voltage ($V_{DS}$), for two p-type GFETs as they were subjected to forward and backward voltage sweeps at a constant back-gate voltage ($V_{BG}$) of 0 V. Each separate curve displayed in Fig. 1d, e represents a different sweep range of $V_{DS}$. In the measurements represented by Fig. 1d, the voltage was swept to a positive maximum, $V_{DSmax}$, ranging from 1 V to 6.5 V in steps of 0.5 V. As demonstrated by the curves, increasing the sweep range appears to increase the hysteresis window of the GFET until $V_{DSmax} = 5$ V, beyond which the direction of hysteresis reverses and the hysteresis window begins to decrease. This phenomenon was also seen in the device characterized in Fig. 1e, wherein each $V_{DSmax}$ value was negative.

Switching occurred at a similar magnitude ($V_{DSmax} = -5.5$ V) despite the difference in polarity with the device seen in Fig. 1d. Hysteresis switching behavior with the increase in $V_{DSmax}$ was taken to be indicative of memristive switching between states of high and low conductance, similar to the switching in oxide-based memristors caused by the initial formation of conductive filaments due to voltage application, known as the forming process[25–27].

To establish the presence of memristive switching mechanisms in GFETs distinct from those seen in traditional oxide-based memristors (i.e., formation/degradation of conductive filaments in the oxide), a series of programming pulses through the back-gate of different GFETs was performed, with results displayed in

Supplementary Note 1. For biases equal to or greater than those utilized in programming through the source/drain, no change in the transfer characteristics or conductance states was noted. This established that no conductive filament formation/degradation was occurring as a result of the back-gate programming pulses, making it clear that the mechanism being utilized to create memory states in GFET memristors is distinct from that utilized in oxide-based memristors. Indeed, as will later be discussed, the bulk oxide ($Al_2O_3$) is not believed to play any major role in the memristive mechanisms shown, with the mechanisms instead being dominated by interactions at the graphene/$Al_2O_3$ interface.

Figure 1f, g, respectively, display the transfer characteristics (i.e., $I_{DS}$ versus $V_{BG}$) for the GFETs at $V_{DS} = 10$ mV, measured immediately following the sweeps represented in Fig. 1d, e that correlate with the given $V_{DSmax}$. As shown in Fig. 1f, sweeping the GFET to a higher positive $V_{DSmax}$ (6.5 V as opposed to 1 V) results in a large shift towards n-type characteristics, with the Dirac point ($V_{Dirac}$) shifting from $V_{Dirac} = 6.4$ V to $V_{Dirac} = -5.8$ V, whereas, as shown in Fig. 1g, sweeping the GFET to a higher negative $V_{DSmax}$ results in a much smaller shift, changing the characteristics of the device from p-type to ambipolar as $V_{Dirac}$ shifts from $V_{Dirac} = 6.5$ V to $V_{Dirac} = -0.2$ V. Hysteresis loops of the drain-to-source current have long been noted in graphene and related materials, including graphene oxide and carbon nanotubes (CNTs). This phenomenon has been the subject of numerous studies and is generally attributed to interactions between the materials and trap sites on their substrates and/or extraneous molecules adsorbed on the material surface or at the material/substrate interface[28,29]. Of these adsorbates, water molecules ($H_2O$) have seen attention in studies due to their prevalence in most ambient environments, as well as due to the use of water baths in traditional graphene transfers[30–32]. While surface-bound $H_2O$ can be easily removed via vacuum or the addition of a passivation layer, $H_2O$ trapped at the graphene/substrate interface requires specific treatments to remove and can have significant impact on the electrical properties of the graphene. An investigation by Cho et al.[33] on the effects of water trapping at the graphene/$Al_2O_3$ interface identified two possible adsorption modes for $H_2O$ trapped at the interface: molecular adsorption (in which the oxygen atom is bound to an $Al_S$ site on the substrate surface) and dissociative adsorption (in which the water molecule is split into an $OH^-$ molecule bound to an $Al_S$ site and a $H^+$ ion bound to an $O_S$ site). The alignment of $H_2O$ relative to graphene differs between the two modes (parallel for $H_2O$ in molecular adsorption and perpendicular for $OH^-$ in dissociative adsorption), leading to differences in the local electrical field, with the field induced by dissociative adsorption being magnitudes larger than that induced by molecular adsorption. The stronger dissociative field, in turn, leads to a higher planar-averaged charge density and p-type doping of the graphene[29,32,33].

Based on the distinctly p-type nature of the GFETs tested and discussed in this paper, it is reasonable to assume that it is a result of dissociative adsorption of $H_2O$ trapped at the graphene/$Al_2O_3$ interface, most likely as a result of the graphene transfer process discussed in the "Methods" section. Similar processes have been noted to result in trapped water adlayers at the interfaces of graphene and a number of different substrates[30,33,34]. Stemming from this, it is also reasonable to assume that the hysteresis shown in Fig. 1d, e is primarily caused by the trapped $H_2O$ as well. To explore this phenomenon further, effort was made to observe the effects of passivation upon the demonstrated GFET hysteresis switching. A separate set of GFETs was fabricated on a separate $Al_2O_3$ substrate and passivated via the deposition of 120 nm of PMMA. Following passivation, the hysteresis switching tests discussed and demonstrated in Fig. 1d, e were performed upon the passivated devices. The results for these tests are

demonstrated in Supplementary Note 2. The hysteresis seen for the positive and negative sweeps in Supplementary Fig. 2a and 2b closely resembles that seen in Fig. 1d, e, respectively. This indicates that the hysteresis and hysteresis switching is not tied to any adsorbates on the free surface of the graphene channel.

However, this does not rule out contributions from adsorbates trapped at the graphene/$Al_2O_3$ interface. Previous studies, such as that by Woong Kim et al.[35], have established that hysteresis due to adsorption of water at the interface can persist following surface passivation. Based on these observations, the forming process discussed in this paper is believed to be a result of switching between different adsorption modes for water molecules trapped at the graphene/$Al_2O_3$ interface. Following fabrication, these molecules are believed to be dissociatively adsorbed on account of the distinctly p-type nature of the transfer characteristics for all GFETs tested, as well as the noticeable hysteresis when observing the swept output characteristics. The increase in the drain bias applied during these sweeps is believed to induce a transition to molecular adsorption of the water molecules. The $OH^-$ molecule and $H^+$ ion bound to an $Al_S$ site and $O_S$ site, respectively, on the $Al_2O_3$ surface would recombine and bind to an $Al_S$ site, with the OH-bonds of the resulting $H_2O$ molecule lying relatively parallel to the plane of the graphene. This is supported by the transition of the GFET transfer characteristics following each sweep; as $V_{DSmax}$ increases in magnitude, $V_{Dirac}$ shifts more and more negative, causing the transfer characteristics to become either ambipolar (for negative bias pulsing) or n-type (for positive bias pulsing). While the GFET is then able to demonstrate analog switching between the n-type and ambipolar states, it is unable to return to the original p-type characteristics indicative of dissociative adsorption. In addition, following the initial bias sweeping, any subsequent sweeping fails to demonstrate any significant hysteresis, a known characteristic of molecular adsorption. This can be seen in Supplementary Fig. 2c and 2d. Further discussion of the hysteresis and the potential contributions of interface defects/adsorbates can be seen in Supplementary Note 3 and 4, respectively. The characteristic switching displayed in Fig. 1f, g indicates the presence of at least two distinct conductance states for GFETs, achievable by applying high source-drain biases across the graphene channel. Since programming time is a vital factor for memory of any form, experiments were performed to observe the time needed to maximize the conductance difference between the conductance states. Figure 1h displays the difference between the conductance of two states as a function of $V_{BG}$ (i.e., the read gate voltage after the sequential application of positive and negative $V_{DS}$ write pulses voltages of magnitude 5 V applied for different pulse durations). A read voltage ($V_{DS} = 10$ mV) was used to extract conductance values following each write pulse, as per the equation $G = \frac{I_{DS}}{V_{DS}}$. A minimal change in the conductance was observed for difference pulse times. Nevertheless, the above experiment demonstrates the ability to program and erase conductance states in graphene simply by applying a $V_{DS}$ pulse of opposite polarity, making it attractive for non-volatile memory (NVM) applications.

A critical qualifier for any NVM is switching endurance, which determines how many times the memory can be overwritten to store new information[27,36–39]. We found that when $V_{DS}$ pulses >6 V are applied to GFETs, the devices experience switching failure after only a few (< 10) cycles. To better analyze the effect of $V_{DS}$ write pulse magnitude on switching endurance, as well as its effect on the memory ratio of GFETs, 200 cycles of positive and negative $V_{DS}$ pulses of different magnitudes were applied to different GFETs. The pulse duration was set to 1 s. A histogram of conductance values following the positive (red) and negative

(blue) pulses extracted from these tests are shown in Fig. 1i. Prior to measurement, each GFET used in this experiment was subjected to sequential positive and negative voltage pulses of magnitude 6 V. This was done to set the device characteristics into the n-type and ambipolar conductance states demonstrated in Fig. 1f, g. Supplementary Note 5 demonstrates that the conductance distributions resulting from write pulses of very large magnitude (>6 V) tend to overlap due to poor cycling endurance. Distributions resulting from write pulses of low magnitude (<4.5 V) also overlap, as these voltage pulses are of insufficient magnitude to induce a threshold shift capable of forming distinct conductance states. Conductance states obtained from positive and negative write pulses of magnitude 5 V, and, to a lesser extent, those obtained from pulses with magnitude 4.5 V, display both a relatively large difference in conductance and a switching endurance >200 cycles. Power consumption for the GFETs is approximately 5 mW for write operations at a pulse magnitude of 5 V and less than 40 nW for read operations at a read voltage of 10 mV. Using a pulse time of 1 s, this establishes a switching energy of approximately 5 mJ.

A major advantage of resistive memory devices is their ability to support multiple memory states, allowing for a single device to encompass multiple bits of memory and therefore possess a higher data storage density. This, in turn, can lead to the development of smaller, more efficient devices, which are highly advantageous for applications such as the Internet of Things (IoT)[24] and mobile devices capable of utilizing ANNs[10]. However, while all memristors are capable of realizing bi-stable (1-bit) memory cells due to their ability to switch between two (ON/OFF) conductance states, there is still significant challenge in implementing memristive devices that can be reliably programmed at a multitude of distinct conductance states[27,40]. Our demonstration of the electrical characteristic switching of GFETs shown in Fig. 1 indicates that the graphene devices can achieve multiple (>2) conductance states and could serve as multi-bit NVM if we can exploit write pulses of different magnitudes. Figure 2a displays the transfer curves for a GFET when negative write voltage pulses of duration 1 s with increasing magnitude are applied to the GFET, starting at 3 V and increasing to 6 V in steps of 0.2 V. The GFET characteristics clearly show a monotonic transition from n-type to ambipolar characteristics. Note that prior to the application of these write pulses, the GFET was set to n-type characteristics via the application of a positive $V_{DS}$ pulse of magnitude 6 V. Clearly, there exists multiple distinct Dirac points between the two end states, resulting in multiple (>2) conductance states for GFETs. For multi-bit memory it is critical to test the retention and distinguishability among the different memory states[27,36,41]. Figure 2b through e display the temporal variation (retention) in the conductance values for each state, measured for a total duration of 100 s, when the GFET is programmed into 2, 4, 8, and 16 conductance states, respectively, through different write pulse step sizes. The read voltage ($V_{DS}$) was kept at 10 mV for all tests shown while $V_{BG}$ was kept at 0 V. Retention and endurance testing over longer durations that what is shown here can be seen in Supplementary Note 6. Accompanying histograms display the conductance distribution for each programming configuration. For each set of states tested, the initial states ($t = 0$ s) were set by applying a −5 V $V_{DS}$ pulse for 1 s. For each subsequent state, pulse time was kept to 1 s in order to maximize the memory ratio between each state. The maximum write pulse magnitude was restricted to ≤5 V in order to ensure high switching endurance, as indicated in Fig. 1g. As evident from Fig. 2b and the corresponding conductance histograms, 2 distinct conductance states with significant memory ratio are achieved by applying $V_{DS}$ write pulses with a step size of 2 V. However, as the number of conductance states increases from 2 states to 4

(Fig. 2c), 8 (Fig. 2d), and 16 (Fig. 2e) states by decreasing write pulse step size to 0.5 V, 0.25 V, and 0.125 V, respectively, the memory ratio diminishes between each state, reducing the distinguishability between the states. We performed similar experiments for positive write pulse polarity, as shown in Fig. 2f. In this case, the GFET was initially set to ambipolar characteristics via the application of a negative $V_{DS}$ pulse of magnitude 6 V. Figure 2f shows that the GFET can be gradually switched to n-type characteristics via the application of positive $V_{DS}$ pulses of increasing magnitude. The distinct Dirac points seen in the process indicate the potential for achieving multiple distinct conductance states. Figure 2g through j display the temporal variation in the conductance values for each state, measured for a total duration of 100 s, when the GFET is programmed into 2, 4, 8, and 16 conductance states, respectively, through different write pulse step sizes. Accompanying histograms display the conductance distribution for each programming configurations. A similar conclusion is drawn regarding the memory ratio and retention for positive write pulse polarity as for negative write pulse polarity. These results indicate the ability to step the conductance states of a single GFET in either direction (i.e., higher conductance to lower or vice versa), as well as the ability to return to previous conductance states by applying voltage pulses of opposite polarity.

It should be called to attention that when operating at ≥16 states, such as what is shown in Fig. 2e, j, the memory ratio between neighboring memory states can decrease significantly to the point where the non-volatility of the devices can be called into question. Indeed, when one considers the accompanying histograms for Fig. 2e, j, it is readily apparent that there is a non-insignificant amount of crossover in the distributions of neighboring conductance (memory) states. While this can negatively affect operation of the GFETs at a higher number of memory states, it does serve to demonstrate the analog ("continuously variable") nature of the conductance states achievable on GFETs. As demonstrated by the other subfigures of Fig. 2, when operating at ≤8 memory states, GFETs maintain non-volatility at the cost of number of memory states. However, they remain able to be programmed to any of the memory states shown in Fig. 2e, j. This provides an attractive amount of flexibility for neuromorphic applications, allowing for the GFETs to achieve targeted conductance (weight) values reliably and accurately as needed. This is exemplified through our demonstration of k-means clustering compared to uniform quantization, as will be discussed in the subsection On-Chip VMM using Graphene Memristors and k-Means Clustering. It should also be noted that for the negative and positive write pulse sequences shown in Fig. 2b through e and Fig. 2g through j, respectively, all tests shown of a given polarity were performed upon the same GFET. A slight difference in the initial $I_{DS}$ values can be seen for testing of both polarities, with a maximum initial current difference of 0.13 μA for the negative pulsing and 0.23 μA for the positive pulsing. These minor differences could be due to shifts in the threshold voltage as a result of varying interface trap state population/depopulation from the high electric field generated during pulsing. While the majority of dangling bonds at the graphene/Al$_2$O$_3$ interface are believed to be occupied by water molecules, there is no doubt a small number still capable of acting as carrier traps. However, based on the consistency of the memory ratios between states and the reduction of hysteresis following the forming process, the overall effect on the GFETs is believed to be minimal.

The conductance switching demonstrated in GFETs following the forming process, as highlighted by Fig. 1d through g, is believed to be the result of dipole moment switching due to the generated electric field. Such effects have been shown to result in

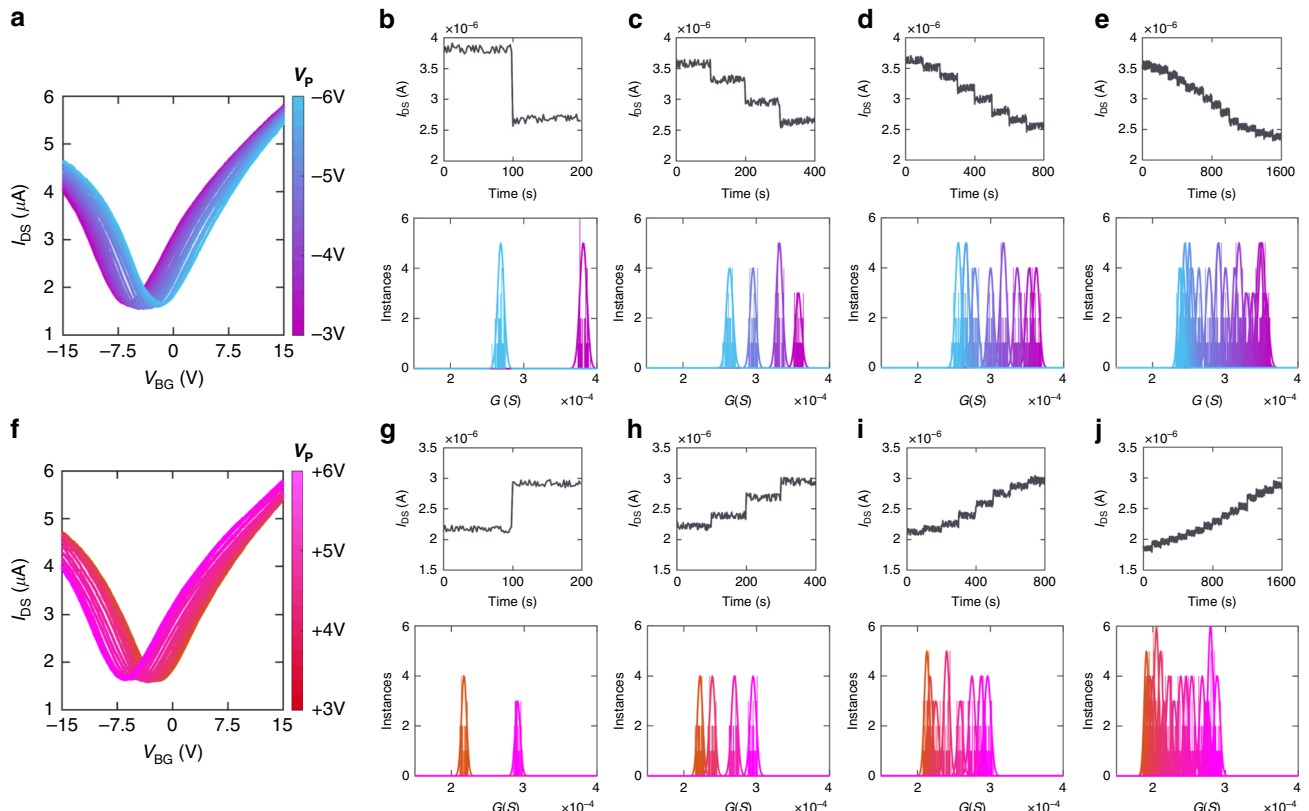

**Fig. 2 Memory levels, memory ratio, and memory retention of graphene memristors. a** Transfer characteristics of a GFET when negative write voltage pulses of duration 1 s with increasing magnitude are applied, starting at −3 V and increasing to −6 V in steps of −0.2 V. The GFET characteristics clearly show a monotonic transition from n-type to ambipolar characteristics. Prior to the application of these write pulses, the GFET was set to n-type characteristics via the application of a positive $V_{DS}$ pulse of magnitude 6 V. Clearly, multiple (>2) memory levels (conductance states) are achieved in graphene memristors. Memory ratio and memory retention, measured for a total duration of 100 s at $V_{BG} = 0$ V, for a graphene memristor programmed into (**b**) 2, (**c**) 4, (**d**) 8, and (**e**) 16 memory levels using different write pulse ($V_{DS}$) step sizes, each of duration 1 s. Accompanying histograms display the conductance distributions for each programming configuration. The maximum write pulse magnitude was restricted to ≤5 V in order to ensure high switching endurance. Significant memory ratio is achieved when $V_{DS}$ step size is 2 V. However, as the number of memory level is increased by decreasing $V_{DS}$ step size to 0.5 V, 0.25 V, and 0.125 V, respectively, the memory ratio diminishes, reducing the distinguishability between the conductance states. **f** Transfer characteristics of an ambipolar GFET when positive write voltage pulses of duration 1 s with increasing magnitude are applied, starting at 3 V and increasing to 6 V in steps of 0.2 V. The GFET returns to n-type characteristics. Memory ratio, memory retention, and corresponding histograms of conductance distributions for the same graphene memristor programmed into (**g**) 2, (**h**) 4, (**i**) 8, and (**j**) 16 memory levels. These results indicate the ability to configure the GFET in precise conductance states, change it in either direction (i.e., higher conductance to lower or vice versa), and return it to previous conductance states by applying voltage pulses of opposite polarity.

threshold and conductance shifting in field effect transistors with interfacial dipole monolayers, leading to the development of distinct memory states[41–45]. Following the forming process (the transition from dissociative adsorption of water molecules at the interface to molecular adsorption), the water molecules are randomly oriented due to the uncoordinated nature of $Al_S$ states at the $Al_2O_3$ surface[33]. In this state, the local electric field generated by the water molecules is far weaker than in dissociative adsorption, owing to the interference caused by the random orientation of neighboring dipoles. As a result, the graphene tends to display ambipolar transfer characteristics as opposed to its initial p-type characteristics. Previous studies have shown that interfacial water molecules at a graphene surface can be reoriented through the application of an external electric field[46,47]. This polarizes the water molecules and can align their dipoles due to their preference for an orientation parallel to the electric field, enhancing the local electric field and increasing conductance of the graphene channel[48]. Experimentally, this phenomenon is reflected by the increase in conductance observed when positive bias pulses are applied to the GFETs through the drain, as demonstrated in Fig. 2g through j. When negative bias

pulses are applied, the water molecules are oppositely polarized, leading to reorientation. This is reflected by the decrease in conductance through negative bias pulsing shown in Fig. 2b through e. The ability for GFETs to switch to and from conductance states without being reset to n-type or ambipolar characteristics could allow for faster writing and erasing of data, as well as higher density data storage, when used as multi-bit memory[49]. Also note that the conductance values corresponding to the different memory states of the GFETs are linearly and symmetrically distributed, fostering high accuracy in ANNs that rely on backpropagation learning rule[50]. Furthermore, the precise control of GFET conductance states can offer tremendous benefit for on-chip training that rely on precise weight updates for faster convergence.

The multi-terminal nature of the GFET-based synaptic device allows for it to be modulated by both the $V_{DS}$ programming pulses mentioned previously and by the $V_{BG}$ applied during read operations. By varying $V_{BG}$, the resistance of the graphene channel can be modulated, allowing for tuning of the conductance states (weight values) programmed into the device via the $V_{DS}$ pulses, as well as the memory ratio between neighboring

states. This tuning of weight values through the application of a separate bias could be considered reminiscent of heterosynaptic plasticity in biological neural networks, in which the stimulation of a given neuron causes a change in the strength (weight) of synaptic connections between other, inactivated, neurons[51,52]. The most well-known example of this mechanism is the existence of modulatory neurons, also known as interneurons. When activated, these neurons release chemicals known as neuro-modulators, differentiated from typical neurotransmitters by their ability to alter synaptic efficacy instead of generating an electrical response. These alterations can last up to several minutes, providing comparatively long-term modulation of synaptic events[53]. In addition, repeated heterosynaptic modulation has, through study, been found to promote the growth/retraction of synaptic connections, creating persistent changes in synaptic weight and contributing to long-term memory formation/storage[54]. This has made the implementation of heterosynaptic plasticity an important goal for developing the next generation of novel neuromorphic systems[52]. The modulation of conductance states afforded by different modulatory bias, $V_{mod}$ ($V_{BG}$), values can be seen in Supplementary Note 7. All measurements were conducted on the same GFET using the same $V_{DS}$ pulsing scheme utilized in Fig. 2e. Each state was held for 100 s with no observable degradation into neighboring states, indicating good retention for all $V_{mod}$. The changes in conductance states and memory ratios between adjacent states as a result of changing $V_{mod}$ indicate the ability to implement synaptic potentiation and depression by using the back-gate bias to increase or decrease conductance states (weight values) independently from the application of programming pulses across the source and drain. Thus, the extra degree of freedom offered by the multiterminal design of GFETs allows for synaptic modeling that is not possible in traditional two-terminal synaptic devices, such as those that operate using oxide-based memristors.

To investigate the scalability of GFET memristors, several sets of GFETs featuring reduced channel lengths (L), ranging from 200 nm to 800 nm in steps of 200 nm, were fabricated on a separate $Al_2O_3$ substrate using the same fabrication processes discussed in the "Methods" section. Half of each set was fabricated with a channel width (W) of 1 µm, while the other half was fabricated with a channel width matching the channel length. The device layout that covered the smallest area while remaining functional was found to be that with L = 400 nm and W = 1 µm, for a total area of 0.4 µm². In addition to the reduced area, the devices demonstrated the ability to shift conductance states at lower pulse magnitudes than the 1 µm channel length (0.5 µm² channel area) devices detailed previously, indicating a channel length/area dependency for the conductance switching mechanism in GFETs.

Supplementary Fig. 8a demonstrates the shifting from initial p-type characteristics indicative of the forming process at lower voltages than was necessary for the 1 µm channel length devices. This can be explained as an effect of the shorter channel length increasing the electric field generated by each pulse, allowing for reorientation of the water molecule dipoles trapped at the graphene/$Al_2O_3$ interface at lower biases. The shift from n-type to ambipolar characteristics when negative low magnitude voltage pulses are applied, as shown in Supplementary Fig. 8b, supports this theory. Interestingly, the 400 nm channel length devices continued to display high endurance even at high pulse magnitudes. Supplementary Fig. 8c and 8d display the shifting of the Dirac point when positive and negative pulses of magnitude 4 V and 5 V, respectively, are applied. Insets show the endurance at each pulse magnitude for 200 cycles. The 5 V pulse cycling, which was noted as the maximum sustainable voltage for the 1 µm channel length devices, shows similar

memory ratio and endurance to that of the 1 µm channel length devices discussed earlier. These results indicate that a wider range of pulse magnitudes can be used for shorter channel devices, potentially increasing the number of distinct achievable memory states while retaining similar endurance. In contrast, all 200 nm channel length devices tested were found to be intrinsically ambipolar and displayed little-to-no shifting when programming pulses were applied. These results were taken to support our initial hypothesis regarding the role of water molecule adsorption and dipole alignment at the interface regarding changing conductance states. It is believed that the small area of the 200 nm channel length devices did not allow for sufficient water molecules to be trapped to demonstrate p-doping via dissociative adsorption or conductance shifting via dipole realignment. The channel scalability demonstrated also shows promise for large-scale integration of GFETs into crossbar-array architectures. As shown and discussed in Supplementary Note 9, the nature of the programming phenomenon of GFETs, bias pulsing through the drain, allows for electrostatic isolation of devices in close proximity to one another despite the presence of a global back-gate. Together with the aforementioned scaling, this indicates the potential for high integration density of GFET memristors, offering an attractive alternative for close-packed memristive device architectures such as dense crossbar-arrays.

**Impact of weight assignment–uniform versus *k*-means clustering**. Weight quantization is inevitable for hardware implementation of ANNs. However, it leads to quantization error since weights must be rounded to the nearest analog value. For ANNs implemented using the traditional von Neumann architecture, weights can be stored in high precision digital memory to reduce the quantization error at the expense of latency, energy inefficiency, and area overhead. On the contrary, ANNs exploiting in-memory computing can suffer from the limited number of analog memory levels offered by the crossbar architecture in spite of high speed, low power, and area efficient design. In this section, we elucidate on the impact of number of analog memory levels on the quantization error for VMM architecture and demonstrate how it can be mitigated by adopting proper quantization techniques. Figure 3a shows a data structure with two vectors, A and B, and their product, C. Vector A contains 5000 matrices of size $1 \times 2$ and vector B contains 5000 matrices of size $2 \times 1$, with matrix elements drawn randomly from some given weight distributions. Figure 3b, c, respectively, show two such weight distributions, namely uniform distribution and normal distribution in the range of $[-1,1]$. Figure 3d shows the schematic of uniform quantization where the data range $[-1, 1]$ is divided into $N$ equally spaced bins, with $N$ being the number of analog memory levels. Any synaptic weight that belongs to a given bin is assigned to the analog memory value associated with that bin. Figure 3e, f, respectively, show the error histogram as a function of $N$ when the weights are drawn from the uniform and the normal distributions corresponding to Fig. 3b, c. The error histogram is computed from ($C_Q - C$), where the elements of vector C, i.e., $c_n$, are the product of matrix $[a_{n1} \ a_{n2}]$ from vector A and matrix $[b_{1n}; b_{2n}]$ from vector B, and the corresponding elements of vector $C_Q$ are the product of quantized $A_Q$ and $B_Q$. Figure 3g shows that the error, as expected, decreases with increasing $N$. However, for similar $N$, the error due to uniform quantization is significantly higher for normally distributed weights when compared to uniformly distributed weights. Since weight distributions in practical scenarios[55–58] are more likely to follow a normal distribution, uniform quantization can lead to significantly high inference inaccuracy. In order to mitigate the challenges associated with uniform quantization, we propose *k*-means clustering based

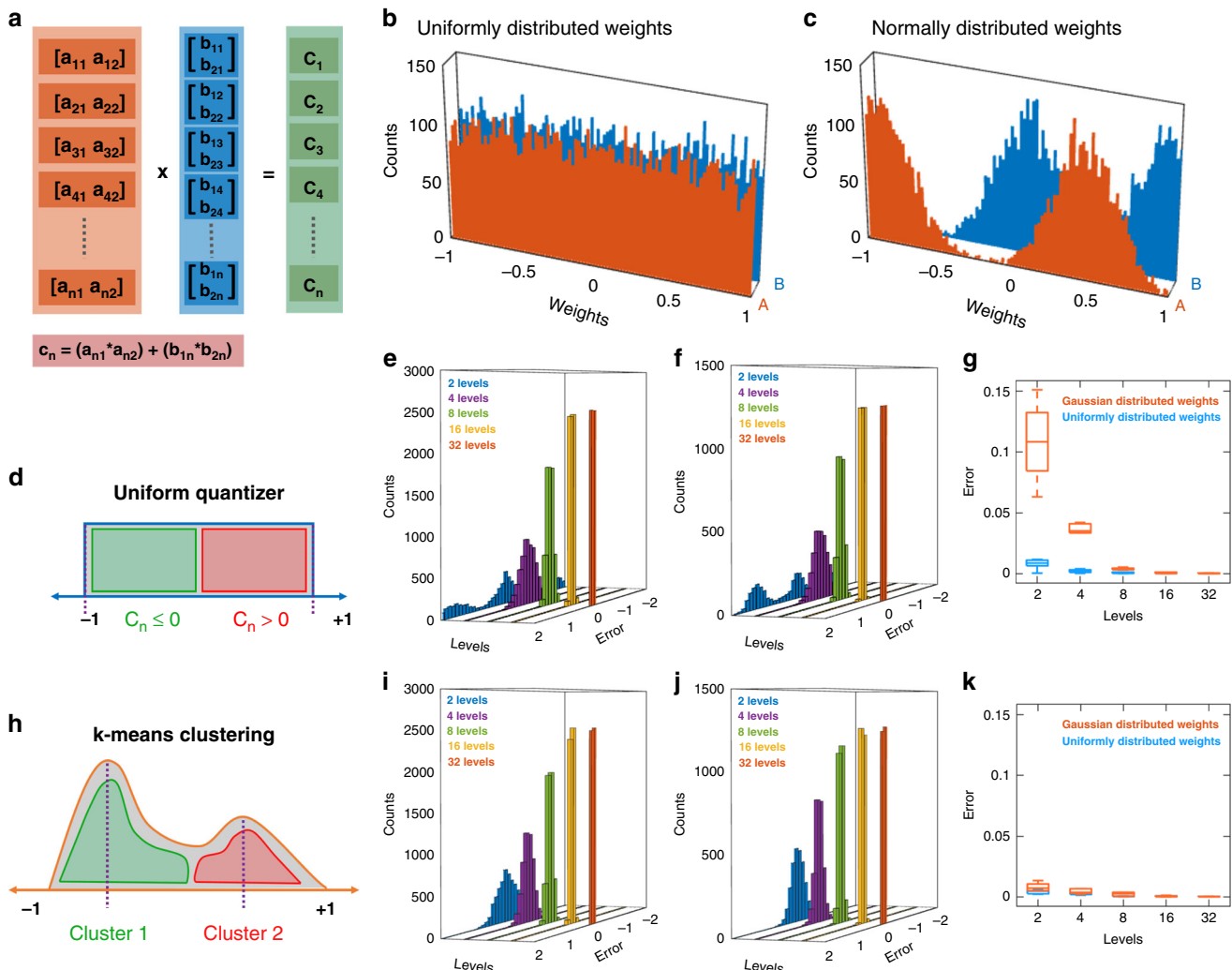

**Fig. 3 Weight assignment using uniform distribution and *k*-means clustering. a** Data structure showing vectors A and B of sizes 1 × 2 and 2 × 1, respectively, and their product, C. Matrix elements for A and B are drawn randomly from (**b**) uniform and (**c**) Gaussian normal weight distributions in the range of [−1, 1]. **d** Uniform quantization where the data range [−1, 1] is divided into *N* equally spaced bins. Any weight that belongs to a given bin is assigned to the analog memory value associated with that bin. The error histogram computed from ($C_Q$ − C), where the elements of $C_Q$ are the product of quantized elements from A and B (i.e., $A_Q$ and $B_Q$) as a function of *N*, when weights are drawn from (**e**) uniform and (**f**) normal distributions, corresponding to (**b**) and (**c**), respectively. **g** Box plot of the error in (**e**) and (**f**), which shows monotonic decrease as *N* increases. Also, for similar *N*, the error is significantly higher for normally distributed weights when compared to uniformly distributed weights. **h** The schematic of *k*-means clustering, which is an unsupervised learning algorithm that divides the *n* data samples into *k* clusters, such that *k* ≤ *n*. The algorithm randomly chooses the centroids, calculates the distance of each point to the centroid, and, finally, minimizes the variance of the distance iteratively to identify the centroids. These centroids are usually located near the mean of the clusters. In *k*-means clustering quantization, weights in a specific cluster are quantized to their respective centroids. The error histogram as a function of *N* when the weights are drawn from the (**i**) uniform and (**j**) normal distributions corresponding to (**b**) and (**c**), respectively. **k** Box plot of the error in (**i**) and (**j**), which shows significant reduction in error for both cases when compared to that shown for uniform quantization in (**g**), especially for normally distributed weights.

quantization. Figure 3h shows the schematic of *k*-means clustering, which is an unsupervised learning algorithm that divides the *n* data samples into *k* clusters, such that *k* ≤ *n*[59]. The algorithm randomly chooses the centroids, calculates the distance of each point to the centroid, and, finally, minimizes the variance of the distance iteratively to identify the centroids. These centroids are usually located near the mean of the clusters. In *k*-means clustering based quantization, weights in a specific cluster are quantized to their respective centroids. Figure 3i, j, respectively, show the error histogram as a function of *N* when the weights are drawn from the uniform and the normal distributions corresponding to Fig. 3b, c and are quantized using *k*-means clustering. As shown in Fig. 3k, the error decreases with increasing *N*. More importantly, *k*-means clustering offers better accuracy compared

to uniform quantization and the benefits are found to be more for normally distributed weights. However, centroids of the weight distributions are not necessarily symmetric or follow linear trends. As such, hardware implementation of *k*-means clustering based quantization will require analog memory not only with multiple levels but also with the capability of configuring the individual memory states. In the following section we experimentally demonstrate this idea based on analog graphene memristive synapses.

**On-chip VMM using graphene memristors and *k*-means clustering**. Figure 4a depicts our graphene-based resistive memory architecture for executing VMM operations (see Supplementary

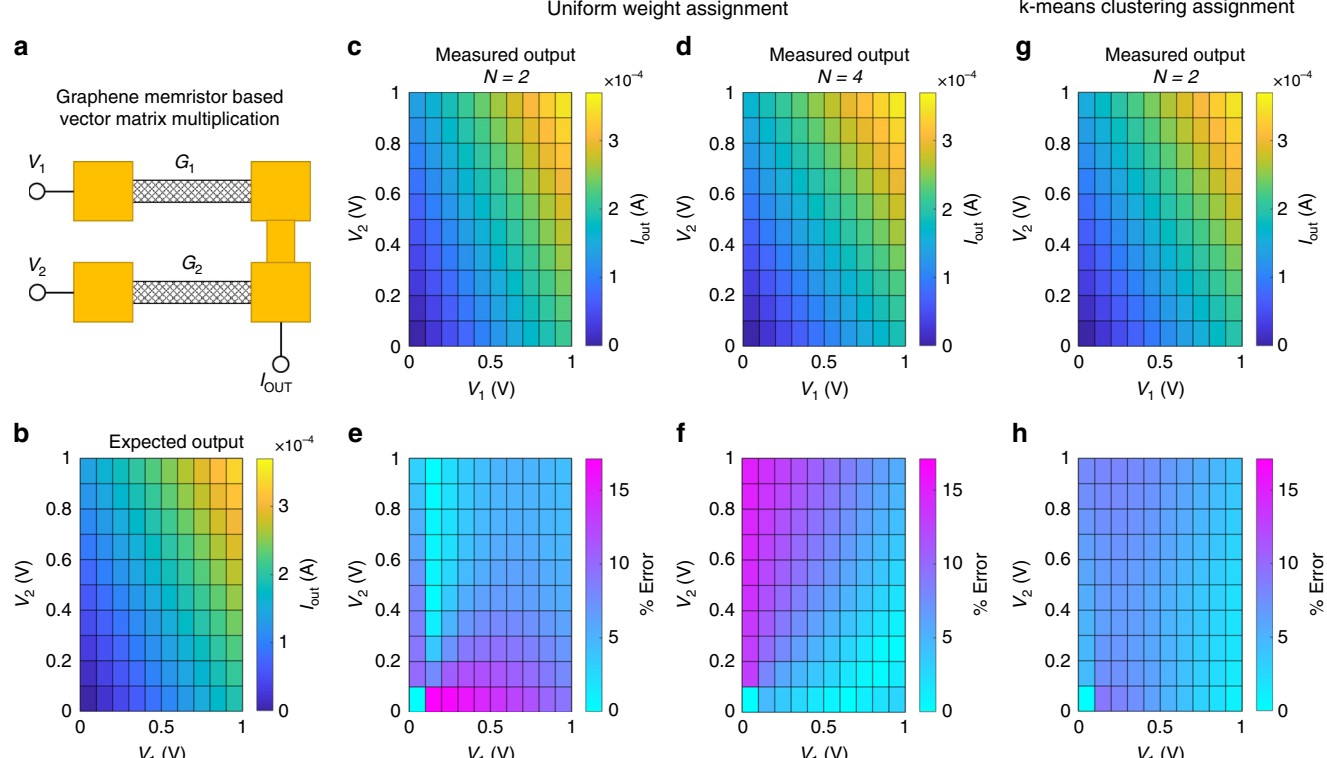

**Fig. 4 Graphene memristor based vector-matrix multiplication (VMM) using k-means clustering. a** Memory architecture for executing VMM operations. Drain voltages ($V_1$ and $V_2$) are used as the input vector and graphene memristor conductance values ($G_1$ and $G_2$) are used as the weight matrix. The output current ($I_{OUT}$) is used as the output vector. **b** Colormap of the expected output current corresponding to different input voltage vectors for $G_1 =$ 215 μS and $G_2 =$ 155 μS. Experimentally obtained output currents when these weights are rounded to the nearest conductance states offered by the respective graphene memristors with uniformly distributed memory levels for (**c**) $N = 2$ and (**d**) $N = 4$. Error between the expected and actual output current for (**e**) $N = 2$ and (**f**) $N = 4$. **g** Experimentally obtained output current and (**h**) error when the weights are rounded to the nearest conductance states following k-means clustering.

Note 9 for programming of successive GFETs in an array). Note that, for any given back-gate voltage ($V_{BG}$), the output current is given by the product of the conductance matrix and input voltage vector following the equation:

$$I_{OUT} = G_1 V_1 + G_2 V_2 = \begin{bmatrix} G_1 & G_2 \end{bmatrix} \begin{bmatrix} V_1 \\ V_2 \end{bmatrix} \quad (1)$$

We consider a situation where the elements of the weight matrix are the centroids of the weight distribution obtained through k-means clustering quantization with $k = 2$. The desired conductance values are, for example, $G_1 = 215$ μS and $G_2 = 155$ μS. Figure 4b shows the colormap of the expected output current corresponding to different input voltage vectors. Figure 4c, d show the experimentally obtained output current when these weights are rounded to the nearest conductance states offered by the respective graphene memristors with uniformly distributed memory levels for $N = 2$ and $N = 4$. For $N = 2$, the allowed conductance states for each GFET are 230 μS and 140 μS, corresponding to the programming voltage pulses of $-3.0$ V and $-5.0$ V, respectively. For $N = 4$, the allowed conductance states for each GFET are 230 μS, 200 μS, 170 μS, and 140 μS corresponding to the programming voltage pulses of $-3.0$ V, $-3.5$ V, $-4.0$ V, and $-5.0$ V, respectively. Figure 4e, f show the error between the expected and actual output current, which is relatively high since the weights can only be rounded to the nearest conductance states $G_1^{U2} = 230$ μS and $G_2^{U2} = 140$ μS for $N = 2$ and to $G_1^{U4} = 200$ μS and $G_2^{U4} = 170$ μS for $N = 4$. Despite the increase in the value of $N$, the colormaps of error in Fig. 4e, f

are similar. This is due to the fact that the programmed conductance values differ from their targeted values by 15 μS for both $N = 2$ and $N = 4$. This means that while the error distribution is shifted due to the change in the relative position of the conductance states, the overall accuracy of the synapse is not improved. This serves to highlight a drawback of uniform weight distribution and its implementation using devices with discrete memory states, such as oxide-based memristors. When the desired weight (conductance) value lies between set states, it can be very difficult for the system to reach it unless it utilizes a very large number of memory states. The analog nature of GFET memristors, on the other hand, allows for precise programming of any weight (conductance) value within the distribution of conductance states. The experimentally obtained output current for when the GFETs are directly programmed to the nearest achievable conductance states, $G_1^K = 214$ μS and $G_2^K = 156$ μS, can be seen in Fig. 4g. The error between this output current and the expected output current shown in Fig. 4b is displayed by Fig. 4h. As would be expected, the error is significantly reduced. See Supplementary Note 10 for the post-programmed characteristics of individual GFETs for all cases. Our demonstration shows the benefits of graphene-based memristors over oxide-based memristors, as the former allows for reliable programming of individual GFETs in an array to specific conductance states. In ANN applications, this allows for the realization of k-means clustering, which improves the computing accuracy. A brief summary and comparison of GFET-based memristive synapses with other 2D material based memristive synapses is provided in a table in Supplementary Note 11.

## Discussion

In conclusion, we have successfully demonstrated graphene-based ultra-thin resistive memory capable of achieving multiple (>16) memory states with necessary retention and endurance. We have also demonstrated that these memory states are configurable to desired conductance values, unlike conventional memristive NVMs, through the application of drain voltage pulses of differing magnitudes. Furthermore, we have discussed and shown through simulation the benefits of $k$-means clustering when compared to uniform quantization as a high precision method for quantizing synaptic weights in ANNs. Finally, we have experimentally demonstrated the ability of analog graphene memristive synapses to allow for the on-chip realization of $k$-means clustering while also establishing the ability to perform VMM operations through the use of conductance states as computational weights. We believe that these results will aid in the development of high precision, low-power, and area-efficient neuromorphic computing engines based on graphene memristors for various incarnations of ANNs.

## Methods

**Device fabrication.** Commercially grown monolayer graphene (Graphenea), procured on copper foil and with PMMA pre-spun, was used in our experiments. A wet transfer method was used to transfer the graphene onto a 1 cm², 50 nm Al$_2$O$_3$ substrate with highly-doped (p++) Si as the back-gate electrode. The PMMA/graphene/Cu-foil stack was placed on a surface of iron (III) chloride (FeCl$_3$), with the graphene-covered side facing away from the solution, in order to wet etch the copper foil. Once the etching was complete, the PMMA/graphene stack was transferred to a deionized (DI) water bath using a glass slide. A total of three DI water baths were used (10 min each), with the glass slide being cleaned with acetone and isopropyl alcohol (IPA) between each transfer. The PMMA/graphene stack was then transferred from the last water bath using the Al$_2$O$_3$ substrate previously mentioned. The substrates were heated for 5 min at 55 °C to evaporate any remaining water on the surface and then heated at 150 °C for 10 h to eliminate any wrinkles that may have originated from the transfer process and promote adhesion between the graphene and substrate. The PMMA was then removed using an acetone bath (10 min), which was followed by an IPA bath (5 min) to clean the sample. The graphene channels were defined using electron-beam (e-beam) lithography (Vistec EBPG5200), with the surrounding graphene being etched using O$_2$ plasma (Vision 320 RIE) at room temperature for 15 s. The source/drain contacts were then defined using e-beam lithography. Ni (40 nm) followed by Au (30 nm) was deposited using e-beam evaporation for the contacts. Prior to each instance of e-beam lithography, photoresists MMA EL6 and PMMA A3 were spun onto the substrate at a rate of 4000 RPM for 40 s, with the MMA EL6 serving to promote adhesion and enhance liftoff of the PMMA A3. Following e-beam lithography, the substrate was developed using a 1:1 mixture of 4-methyl-2-pentanone (MIBK) and pure IPA followed by pure IPA for 60 s and 45 s, respectively. Liftoff was performed by submerging the substrate in acetone for approximately 10 min, with a subsequent IPA bath of approximately 5 min to clean the substrate of any residue. All devices were fabricated with a dual-channel structure possessing channel lengths of 1 μm and channel widths of 0.5 μm.

**Device measurements.** Electrical characterization was performed at room temperature in high vacuum (≈10⁻⁶ Torr) on a Lake Shore CRX-VF probe station and using a Keysight B1500A parameter analyzer.

## Data availability

The datasets generated during and/or analyzed during the current study are available from the corresponding authors on reasonable request.

## Code availability

The codes used for plotting the data are available from the corresponding authors on reasonable request.

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

## Acknowledgements
This work was supported by the Army Research Office (ARO) through Contract Number W911NF1920338.

## Author contributions
S.D. and T.F.S. conceived the idea and designed the experiments. T.F.S. performed the experiments. A.O. performed the kNN simulations. All the authors analyzed the data, discussed the results, agreed on their implications, and contributed to the preparation of the manuscript.

## Competing interests
The authors declare no competing interests.
