## [Peer Review File · Nature Communications]

REVIEWER COMMENTS

Reviewer #1 (Remarks to the Author):

The paper by Schranghamer et al. entitled "Graphene Memristive Synapses for High Precision Neuromorphic Computing" demonstrated graphene-based atomically thin memristive synapses using traditional GFET device. The authors fabricated the back-gate GFET synapse device using CVD grown graphene as a channel and Al₂O₃ as gate dielectrics. Their graphene-based memristive synapse showing non-volatile and multi-bit characteristics through the application of drain voltage pulses of differing magnitudes. The authors also simulated "on-chip vector matrix multiplication (VMM)" using Graphene memristor.

The results are convincing but I am concerned with some aspect that are reported in the following:

1) Underlining hysteresis switching mechanism was not explained to support experimental data in Figure 1(d) and (e). Authors claim that the switching mechanism in GFET synapse is similar oxide-based memristors, if so then fruitful explanation on effect of interactions between defects in graphene layer, oxide ions and interfaces are highly desirable. The switching behavior shown in Figure 1(d) and (e) are very common in unpassivated GFET devices, So I think authors should perform the measurement on passivated graphene channel. Why authors prefer to perform the "write" operation on channel instead on gate? I believe that atomically thin graphene may not show the memristive effect if the "write" operation performed vertically through gate.

2) Authors claim the multi-level (>16) and non-volatile graphene (GFET) based synapses with desirable retention and endurance as shown in Figure 2(e) and (j). But from Figure 2(e) and (j) it seems that memory states are unstable (volatile state) and became saturated after certain write pulse (VDS) steps (Figure 2(e)), Please make comment on it. Also, why the initial channel current level for different write pulse steps (2V, 0.5V, 0.25 and 0.125V) are different (as shown in Figure 2b-2j)? Do the authors use different devices for those measurements? In supporting 2, the Dirac point in pristine devices (GFET 3 and GFET 4) are shifted toward positive / negative side, which may indicate PMMA doping or something else. Please make comment on it. In general, the Dirac point of pristine undoped graphene is expected to be located around zero gate voltage

3) The integration density of the synapse is of primary importance in order to construct a functional neuromorphic device. In most of graphene base devices notable challenges such as device scalability and large-scale device integration remain as big challenge. Within the paper the scaling and integration capabilities of the proposed device are not specified. I think authors should add some explanations on this aspect in revised manuscript.

4) Several studies have been proposed on 2D material based memristor synapse such as graphene barristor synapse and graphene ferroelectric synapse. It is an important to discuss those studies and compare their performance with each other (especially in terms of operation speed, multilevel memory states and endurance/ retention. Short summary of previous studies in the form of table is highly recommended.

Reviewer #2 (Remarks to the Author):

The major claim of the paper is the experimental demonstration of a non-volatile graphene-based resistive memory device 16 conductance states. They say that non-volatile graphene memory is not new with 2 memory states (1-bit) on a single device. The authors show that the graphene memristive synapses possess desirable retention and switching endurance with an enhanced accuracy with respect to other synaptic devices. Moreover, they demonstrate the possibility to realize multi-bit and non-volatile graphene memristive synapses for the realization of area and energy efficient hardware for neuromorphic computing and for Internet of Things (IoT).

The reported results have some elements of novelty and interest from the technological and device point of view. The conclusions are original and the reported references are complete.

However, due to the nature of the results and of the scientific activities I think that the paper is more suitable for a journal focused on devices and technology.

Reviewer #3 (Remarks to the Author):

In this manuscript, the authors demonstrated a non-volatile graphene-based memristive device with multiple conductance states, desirable retention and switching endurance, which allowed for the hardware implementation of quantization through K-means clustering and resulted in enhanced accuracy when compared to the uniform weight quantization used by other synaptic devices. The manuscript was generally well organized and well written. Supporting information was relatively in a good match with the explanations. However, before the acceptance of this manuscript, the following issues shall be addressed.

(1) The authors did not make it clear why they utilized three-terminal graphene field effect transistors to implement memristive behavior. What are the advantages of the material and the structure when compared to the traditional oxide-based memristors?

(2) The direction of hysteresis and the hysteresis window of some curves in Fig. 1d and 1e are unclear and clear figures should be provided.

(3) The authors claimed that increasing the sweep range appeared to increase the hysteresis window of the GFET until $V_{DSmax} = 5\text{ V}$, after which the direction of hysteresis reversed and the hysteresis window began to decrease, which was similar to the situation in opposite voltage polarity. Why the direction of hysteresis reversed? And why did the hysteresis window gradually increase initially and then decrease?

(4) Are there any defects at the interface between alumina and graphene? If the defects exist, will they affect the channel conductance when the gate voltage is applied?

(5) As shown in Fig. 1f and 1g, why higher positive and negative source-drain voltages can induce typical n-type and ambipolar characteristics respectively? The internal mechanism needs to be explained in detail and experimentally verified.

(6) The operating voltage of 4.5 V is still relatively large for memristive devices and a switching endurance more than 200 cycles is also insufficient for neuromorphic computing. How to further decrease the voltage and improve the endurance ability?

(7) What's the energy consumption of graphene field effect transistors?

(8) There seems to be no obvious error difference in Fig. 4e and 4f when N increase from 2 to 4. Please make more comments.

(9) There are some format mistakes that need to be dealt with in the citation.

Response to Reviewers

Reviewer 1:

The paper by Schranghamer et al. entitled “Graphene Memristive Synapses for High Precision Neuromorphic Computing” demonstrated graphene-based atomically thin memristive synapses using traditional GFET device. The authors fabricated the back-gate GFET synapse device using CVD grown graphene as a channel and Al₂O₃ as gate dielectrics. Their graphene-based memristive synapse showing non-volatile and multi-bit characteristics through the application of drain voltage pulses of differing magnitudes. The authors also simulated “on-chip vector matrix multiplication (VMM)” using Graphene memristor.

The results are convincing but I am concerned with some aspect that are reported in the following:

1) Underlining hysteresis switching mechanism was not explained to support experimental data in Figure 1(d) and (e). Authors claim that the switching mechanism in GFET synapse is similar oxide-based memristors, if so then fruitful explanation on effect of interactions between defects in graphene layer, oxide ions and interfaces are highly desirable. The switching behavior shown in Figure 1(d) and (e) are very common in unpassivated GFET devices, So I think authors should perform the measurement on passivated graphene channel. Why authors prefer to perform the “write” operation on channel instead on gate? I believe that atomically thin graphene may not show the memristive effect if the "write" operation performed vertically through gate.

We are glad that the reviewer found the results convincing. The reviewer has raised some excellent points related to the work. We completely agree with the reviewer that a clearer and more informative explanation of the hysteresis switching mechanism should be included in our manuscript. In addition, we acknowledge that the statement “programmable conductance in graphene field effect transistors (GFETs) devices similar to that seen in oxide-based memristors” was somewhat misleading but was not intended to insinuate that GFET memristors share the same programming mechanism as oxide-based memristors (i.e. formation/degradation of conductive filaments in the oxide). We also concur that testing passivated GFETs should allow for a clearer analysis of the switching mechanisms involved.

We have added the following brief discussion to address the distinction between oxide-based memristive mechanisms and that shown by GFETs in the revised manuscript: To establish the presence of memristive switching mechanisms in GFETs distinct from those seen in traditional oxide-based memristors (i.e. formation/degradation of conductive filaments in the oxide), a series of programming pulses through the back-gate of different GFETs was performed, with results displayed in **Supporting Information 1** (included here as Fig. R1). For biases equal to or greater than those utilized in programming through the source/drain, no change in the transfer characteristics or conductance states was noted. This established that no conductive filament formation/degradation was occurring as a result of the programming pulses, making it clear that the mechanism being utilized to create memory states in GFET memristors is distinct from that utilized in oxide-based memristors. Indeed, as will later be discussed, the bulk oxide (Al_2O_3) is not believed to play any major role in the memristive mechanisms shown, instead being dominated by interactions at the graphene/ Al_2O_3 interface.

Figure R1: GFET schematic and programming terminal. a.) Schematic of GFETs consisting of a monolayer graphene channel and floating back-gate stack. Electrodes capable of supporting programming biases are identified. b.) V_{DS} programming results for positive and negative pulsing of magnitude 6 V and width 1 s. c.) V_{BG} programming results for the same pulsing scheme as in (b). Despite the large biases applied to the back-gate, no noticeable change in the transfer characteristics was seen, establishing that programming of GFETs is limited to bias pulses applied through the source/drain and thus is not a result of conductive filament formation/degradation like oxide-based memristors.

We have added the following discussions on the hysteresis and conductance switching mechanisms to the revised manuscript: Hysteresis loops of the drain-to-source current have long been noted in graphene and related materials, including graphene oxide and carbon nanotubes

(CNTs). This phenomenon has been the subject of numerous studies and is generally attributed to interactions between the materials and trap sites on their substrates and/or extraneous molecules adsorbed on the material surface or at the material/substrate interface [1, 2]. Of these adsorbates, water molecules (H_2O) have seen attention in studies due to their prevalence in most ambient environments, as well as the use of water baths in traditional graphene transfers [3-5]. While surface-bound H_2O can be easily removed via vacuum or the addition of a passivation layer, H_2O trapped at the graphene/substrate interface requires specific treatments to remove and can have significant impact on the electrical properties of the graphene. An investigation by Cho *et al* [6] on the effects of water trapping at the graphene/ Al_2O_3 interface identified two possible adsorption modes for water trapped at the interface: molecular adsorption (in which the oxygen atom is bound to an Al_s site on the substrate surface) and dissociative adsorption (in which the water molecule is split into an OH^- molecule bound to an Al_s site and a H^+ ion bound to an O_s site). The alignment of H_2O relative to graphene differs between the two modes (parallel for H_2O in molecular adsorption and perpendicular for OH^- in dissociative adsorption), leading to differences in the local electrical field, with the field induced by dissociative adsorption being magnitudes larger than that induced by molecular adsorption. The stronger dissociative field in turn leads to a higher planar-averaged charge density and p-type doping of the graphene [2, 5, 6].

Based on the distinctly p-type nature of the GFETs tested and discussed in this manuscript, it is reasonable to assume that it is a result of dissociative adsorption of H_2O trapped at the graphene/ Al_2O_3 , most likely as a result of the graphene transfer process discussed in the *Methods* section. Similar processes have been noted to result in trapped water adlayers at the interfaces of graphene and a number of different substrates [3, 6, 7]. Stemming from this, it is also reasonable to assume that the hysteresis shown in Fig. 1d and 1e is primarily caused by the trapped H_2O as well. To explore this phenomenon further, effort was made to observe the effects of passivation upon the demonstrated GFET hysteresis switching, a separate set of GFETs was fabricated on a separate Al_2O_3 substrate and passivated *via* the deposition of 120 nm of PMMA. Following passivation, the hysteresis switching tests discussed and demonstrated in Fig. 1d and 1e were performed upon the passivated devices. The results for these tests are demonstrated in ***Supporting Information 2*** (included here as Fig. R2). The hysteresis seen for the positive and negative sweeps in ***Supporting Information 2a*** and ***2b*** closely resembles that seen in Fig. 1d and 1e, respectively.

This indicates that the hysteresis and hysteresis switching is not tied to any adsorbates on the free surface of the graphene channel.

Figure R2: Hysteresis of PMMA passivated graphene devices. Output characteristics of as-fabricated GFETs following PMMA passivation at a back-gate bias of $V_{BG} = 0$ V for different V_{DS} sweep ranges denoted by V_{DSmax} from a) 1 V to 6.5 V and b) -1 V to -6.5 V in steps of 0.5 V. The arrows denote the sweep direction, with blue representing the forward sweep from 0 V and black representing the backward sweep from V_{DSmax} . Results are similar to those for unpassivated GFETs. As with the output characteristics of unpassivated GFETs shown in Fig. 1d and 1e, hysteresis window initially increases with increasing V_{DSmax} and then reverses direction and starts to decrease. The presence of a passivation layer on the graphene channel establishes that the hysteresis switching, and thus the forming mechanism, does not rely on adsorbates (oxygen molecules, water molecules, etc.) on the graphene free surface. Note that this does not rule out interactions with adsorbates trapped at the graphene/ Al_2O_3 interface. V_{DS} sweeping was repeated following that shown in (a) and (b), with the results being displayed in (c) and (d), respectively. Little-to-no hysteresis was seen, indicating a distinct state change in the GFETs, and confirming the existence of the forming process. Due to the passivation layer, it can be assumed that the mechanisms are thus dominated by interactions at the graphene/ Al_2O_3 interface.

However, this does not rule out contributions from adsorbates trapped at the graphene/ Al_2O_3 interface. Previous studies, such as that by Woong Kim *et al* [8], have established that hysteresis due to adsorption of water at the interface can persist following surface passivation. Based on these observations, the forming process discussed in this paper is believed to be a result of switching between different adsorption modes for water molecules trapped at the graphene/ Al_2O_3 interface. Following fabrication, these molecules are believed to be dissociatively adsorbed on account of

the distinctly p-type nature of the transfer characteristics for all GFETs tested as well as the noticeable hysteresis when observing the swept output characteristics. The increase in the drain bias applied during these sweeps is believed to induce a transition to molecular adsorption of the water molecules. The OH⁻ molecule and H⁺ ion bound to an Al_S site and O_S site, respectively, on the Al₂O₃ surface would recombine and bind to an Al_S site, with the OH-bonds of the resulting H₂O molecule lying relatively parallel to the plane of the graphene. This is supported by the transition of the GFET transfer characteristics following each sweep; as V_{DSmax} increases in magnitude, V_{Dirac} shifts more and more negative, causing the transfer characteristics to become either ambipolar (for negative bias pulsing) or n-type (for positive bias pulsing). While the GFET is then able to demonstrate analog switching between the n-type and ambipolar states, it is unable to return to the original p-type characteristics indicative of dissociative adsorption. In addition, following the initial bias sweeping, any subsequent sweeping fails to demonstrate any significant hysteresis, a known characteristic of molecular adsorption. This can be seen in Fig. R2c and R2d.

The conductance switching demonstrated in GFETs following the forming process, as highlighted by Fig. 1d through 1g, is believed to be the result of dipole moment switching due to the generated electric field. Such effects have been shown to result in threshold and conductance shifting in field effect transistors with interfacial dipole monolayers, leading to the development of distinct memory states [9-13]. Following the forming process (the transition from dissociative adsorption of water molecules at the interface to molecular adsorption), the water molecules are randomly oriented due to the uncoordinated nature of Al_S states at the Al₂O₃ surface [6]. In this state, the local electric field generated by the water molecules is far weaker than in dissociative adsorption, owing to the interference caused by the random orientation of neighboring dipoles. As a result, the graphene tends to display ambipolar transfer characteristics as opposed to its initial p-type characteristics. Previous studies have shown that interfacial water molecules at a graphene surface can be reoriented through the application of an external electric field [14, 15]. This polarizes the water molecules and can align their dipoles due to their preference for an orientation parallel to the electric field, enhancing the local electric field and increasing conductance of the graphene channel [16]. Experimentally, this phenomenon is reflected by the increase in conductance observed when positive bias pulses are applied to the GFETs through the drain, as demonstrated in Fig. 2g through 2j. When negative bias pulses are applied, the water molecules are oppositely

polarized leading to reorientation. This is reflected by the decrease in conductance through negative bias pulsing shown in Fig. 2b through 2e.

2) Authors claim the multi-level (>16) and non-volatile graphene (GFET) based synapses with desirable retention and endurance as shown in Figure 2(e) and (j). But from Figure 2(e) and (j) it seems that memory states are unstable (volatile state) and became saturated after certain write pulse (VDS) steps (Figure 2(e)), Please make comment on it. Also, why the initial channel current level for different write pulse steps (2V, 0.5V, 0.25 and 0.125V) are different (as shown in Figure 2b-2j)? Do the authors use different devices for those measurements? In supporting 2, the Dirac point in pristine devices (GFET 3 and GFET 4) are shifted toward positive / negative side, which may indicate PMMA doping or something else. Please make comment on it. In general, the Dirac point of pristine undoped graphene is expected to be located around zero gate voltage.

We agree with the reviewer's statement regarding the Fig. 2e and 2j, and do not dispute that when operating at ≥ 16 states the memory ratio between neighboring memory states can decrease significantly to the point where the non-volatility of the devices can be called into question. The reviewer also has reasonable concerns regarding the initial current levels for the experiments shown in Fig. 2 and the status of the devices shown in **Supplementary Information 2**.

We have added the following discussion on the volatility and saturation of memory states to the revised manuscript: It should be called to attention that when operating at ≥ 16 states, such as what is shown in Fig. 2e and 2j, the memory ratio between neighboring memory states can decrease significantly to the point where the non-volatility of the devices can be called into question. Indeed, when one considers the accompanying histograms for Fig. 2e and 2j, it is readily apparent that there is a non-insignificant amount of crossover in the distributions of neighboring conductance (memory) states. While this can negatively affect operation of the GFETs at a higher number of memory states, it does serve to demonstrate the analog ("continuously variable") nature of the conductance states achievable on GFETs. As demonstrated by the other subfigures of Fig. 2, when operating at ≤ 8 memory states, GFETs maintain non-volatility at the cost of number of memory states. However, they remain able to be programmed to any of the memory states shown in Fig. 2e and 2j. This provides an attractive amount of flexibility for neuromorphic applications,

allowing for the GFETs to achieve targeted conductance (weight) values reliably and accurately as needed. This is exemplified through our demonstration of k-means clustering compared to uniform quantization, as will be discussed in the section *On-Chip Vector Matrix Multiplication (VMM) using Graphene Memristors and k-Means Clustering*.

We have added the following statement regarding the differences in the initial current shown in Fig. 2 to the revised manuscript: It should also be noted that the negative and positive write pulse sequences shown in Fig 2b through 2e and Fig 2g through 2j, respectively, all tests shown of a given polarity were performed upon the same GFET. A slight difference in the initial I_{DS} values can be seen for testing of both polarities, with a maximum initial current difference of 0.13 μA for the negative pulsing and 0.23 μA for the positive pulsing. These minor differences could be due to shifts in the threshold voltage as a result of varying interface trap state population/depopulation from the high electric field generated during pulsing. While the majority of dangling bonds at the graphene/ Al_2O_3 interface are believed to be occupied by water molecules, there is no doubt a small number still capable of acting as carrier traps. However, based on the consistency of the memory ratios between states and the reduction of hysteresis following the forming process, the overall effect on the GFETs is believed to be minimal.

We have added the following discussion on possible doping effects for the GFETs shown in Supporting Information 2 (now referred to as Supporting Information 7) to the revised supplemental materials: A discrepancy that should be brought to attention is the slight negative and positive shifts in V_{Dirac} seen in the post-forming characteristics for GFET 3 and GFET 4, respectively. For the devices in question, it is possible that the slight positive/negative shifts of V_{Dirac} are due to the presence of non-water adsorbates, such as resist residue. For GFET 4, the slight shift of V_{Dirac} towards 0 V seen during the experiments where the device was not programmed suggests that the initial offset may have been due to incomplete population/depopulation of trap states at the graphene/ Al_2O_3 interface, possibly from the water molecule adlayer, that was remedied by subsequent sweeping of the gate voltage.

3) The integration density of the synapse is of primary importance in order to construct a functional neuromorphic device. In most of graphene base devices notable challenges such as

device scalability and large-scale device integration remain as big challenge. Within the paper the scaling and integration capabilities of the proposed device are not specified. I think authors should add some explanations on this aspect in revised manuscript.

This is an excellent suggestion by the reviewer.

We have added the following discussion on device scaling and integration capabilities to the revised manuscript: To investigate the scalability of GFET memristors, several sets of GFETs featuring reduced channel lengths (L), ranging from 200 nm to 800 nm in steps of 200 nm, were fabricated on a separate Al₂O₃ substrate using the same fabrication processes discussed in the *Methods* section. Half of each set was fabricated with a channel width (W) of 1 μm, matching that of the originally tested devices, while the other half was fabricated with a channel width matching the channel length. The device layout that covered the smallest area while remaining functional was found to be that with L = 400 nm and W = 1 μm, for a total area of 0.4 μm². In addition to the reduced area, the devices demonstrated the ability to shift conductance states at lower pulse magnitudes than the 1 μm² devices detailed previously, indicating a channel length/area dependency for the conductance switching mechanism in GFETs. ***Supporting Information 6*** (included here as Fig. R3) demonstrates the shifting from initial p-type characteristics indicative of the forming process at lower voltages than was necessary for the 1 μm² devices. This can be explained as an effect of the shorter channel length increasing the electric field generated by each pulse, allowing for reorientation of the water molecule dipoles trapped at the graphene/Al₂O₃ interface at lower biases. The shift from n-type to ambipolar characteristics when negative low voltage pulses are applied, as shown in ***Supporting Information 6b***, supports this theory. Interestingly, the 400 nm channel length devices continued to display high endurance even at high pulse magnitudes. ***Supporting Information 6c*** and ***6d*** display the shifting of the Dirac point when positive and negative pulses of magnitude 4 V and 5 V, respectively, are applied. Insets show the endurance at each pulse magnitude for 200 cycles. The 5 V pulse cycling, which was noted as the maximum sustainable voltage for the 1 μm channel length devices, shows similar memory ratio and endurance to that of the 1 μm channel length devices discussed earlier. These results indicate that a wider range of pulse magnitudes can be used for shorter channel devices, potentially increasing the number of distinct achievable memory states while retaining similar endurance. In

Figure R3: Forming, Conductance Switching, and Endurance of Scaled GFET. a) Transfer characteristics of a p-type GFET with channel length 400 nm and channel width 1 μm when positive write voltage pulses of duration 1 s with increasing magnitude are applied. Starting at +1 V and increasing to +3.4 V in steps of +0.2 V. The transition from p-type to n-type characteristics shown is similar to that demonstrated by the forming process displayed in Fig. 1, indicating the presence of memristive switching in scaled GFETs. The lower voltages needed to achieve forming for the scaled GFET indicates strong channel length/area dependence of the forming mechanism, implying that it is electric field dominated. b) Transfer characteristics of the same GFET when negative write pulses of the same magnitudes and step sizes as (a) are applied. The noticeable shift to more ambipolar characteristics from the n-type characteristics programmed in (a) confirms memristive switching in scaled GFETs. c-d) Transfer characteristics of the GFET for +/- c) 4 V and d) 5 V switching. Despite shifting beginning at lower voltages, scaled GFET demonstrates the ability to be programmed up to the maximum voltages established for longer channel devices, indicating that shorter channel devices may be able to utilize a wider distribution of conductance states (memory levels). Insets display endurance testing over 200 cycles for respective pulse magnitudes with no observable degradation. V_{Dirac} shifting and endurance results of (c) and (d) are comparable to those of longer channel devices, indicating that GFETs are able to be scaled without adverse effects on device characteristics.

contrast, all 200 nm channel length devices tested were found to be intrinsically ambipolar and displayed little-to-no shifting when programming pulses were applied. These results were taken to support our initial hypothesis regarding the role of water molecule adsorption and dipole alignment at the interface regarding changing conductance states. It is believed that the small area of the 200 nm channel length devices did not allow for sufficient water molecules to be trapped to

demonstrate p-doping via dissociative adsorption or conductance shifting via dipole realignment. The channel scalability demonstrated also shows promise for large-scale integration of GFETs into crossbar-array architectures. As shown and discussed in **Supporting Information 7**, the nature of the programming phenomenon of GFETs, bias pulsing through the drain, allows for electrostatic isolation of devices in close proximity to one another despite the presence of a global back-gate. Together with the aforementioned scaling, this indicates the potential for high integration density of GFET memristors, offering an attractive alternative for close-packed memristive device architectures such as dense crossbar-arrays.

4) Several studies have been proposed on 2D material based memristor synapse such as graphene barristor synapse and graphene ferroelectric synapse. It is an important to discuss those studies and compare their performance with each other (especially in terms of operation speed, multilevel memory states and endurance/ retention). Short summary of previous studies in the form of table is highly recommended.

We would like to thank the reviewer for their comment and have already included a table briefly comparing the memory characteristics on previous demonstrations of 2D materials-based synapses. **Supporting Information 9** (included here as *Table 1*) briefly summarizes and compares the results of the proposed GFET memristive synapse to previous memristive synapses on 2D materials in a chronological order. More in-depth reviews can be found using Ref. 33 through 35.

Authors	Device Structure	# of Memory States	Memory Ratio	Memory Retention	Memory Endurance (#) of Cycles	Operating Speed	Switching Power/Energy
Choi, Min Sup, et al. (2013) [17]	MoS ₂ /Graphene Heterostructure	2	~ 1E4	1400 s	> 110	100 μs	-
Choi, Min Sup, et al. (2013) [17]	Graphene/MoS ₂ Heterostructure	2	~ 2	1200 s	> 100	1 ms	-
Tian, He, et al. (2017) [18]	2D (PEA) ₂ PbBr ₄ Perovskite	4	1E2	1000 s	100	10 ms	400 fJ

Shi, Y., et al. (2017) [19]	Au/Ti/ h -BN/Cu Electronic Synapse	2	1E2- 1E4	-	-	20-200 μ s	-
Sharbati, Mohammad Taghi, et al. (2018) [20]	Graphene/Li+ Electrochemical Synapse	250	7	13 hrs	> 500	10 ms	< 500 fJ
Sangwan, Vinod K., et al. (2018) [21]	Polycrystalline Monolayer MoS ₂ FET	2	1E2	24 hrs	475	1 ms	-
Zhu, Jiadi, et al. (2018) (2018) [22]	Ion-gated WSe ₂ FET	-	~ 1E5	5000 s	-	100 μ s	30 fJ
Li, Da, et al. (2018) [23]	Multi-layer MoS ₂ FET	2	-	-	6000	2-5 ms	-
Shi, Yuanyuan, et al. (2018) [24]	Multi-layer h -BN Synapse	2	25-1E4		> 1000	10 ns	6000 pW
Huh, Woong, et al. (2018) [25]	Vertically Integrated WO _{3-x} memristor and WSe ₂ /Graphene barristor	4	1E5	1000 s	1000	10 ms	0.1 nW
Zhu, Xiaojian, et al. (2019) [26]	Li _x MoS ₂ Device	2	> 100	8000 s	1000	1 ms	-
He, Congli, et al. (2020) [27]	MoS ₂ Multiterminal FET	6	1E5	> 1000 s	> 100	50 ns	7.3 fJ
(This work)	Graphene FET	> 16	~ 12	> 1000 s	500	1 s	5 mJ

Reviewer 2:

The major claim of the paper is the experimental demonstration of a non-volatile graphene-based resistive memory device 16 conductance states. They say that non-volatile graphene memory is not new with 2 memory states (1-bit) on a single device. The authors show that the graphene memristive synapses possess desirable retention and switching endurance with an enhanced accuracy with respect to other synaptic devices. Moreover, they demonstrate the possibility to realize multi-bit and non-volatile graphene memristive synapses for the realization of area and energy efficient hardware for neuromorphic computing and for Internet of Things (IoT). The reported results have some elements of novelty and interest from the technological and device point of view. The conclusions are original and the reported references are complete.

However, due to the nature of the results and of the scientific activities I think that the paper is more suitable for a journal focused on devices and technology.

We are glad that the reviewer found the contents of the manuscript to be novel and interesting. However, we respectfully disagree with the reviewer regarding the suitability of the manuscript for *Nature Communications*. We do not dispute that the manuscript predominantly focuses on discussing the results obtained for our investigation into graphene-based resistive memory. However, the reasons behind this investigation (the slow decline of the von Neumann architecture and subsequent push for biologically-inspired computing) have wide-reaching, multi-disciplinary connections that extend our work far outside of just being an investigation into a novel device technology. This is touched on in the manuscript by our investigation into device-level vector matrix multiplication (VMM) operations using GFETs, where we establish that not only are GFETs capable of storing the weight values necessary for VMM as conductance states but that they are also capable of realizing advanced quantization techniques such as k-means clustering at a device level. This, in turn, establishes the suitability of GFETs to be utilized as analog synapses for neuromorphic computing and artificial neural networks.

Reviewer 3:

In this manuscript, the authors demonstrated a non-volatile graphene-based memristive device with multiple conductance states, desirable retention and switching endurance, which allowed for the hardware implementation of quantization through K-means clustering and resulted in enhanced accuracy when compared to the uniform weight quantization used by other synaptic devices. The manuscript was generally well organized and well written. Supporting information was relatively in a good match with the explanations. However, before the acceptance of this manuscript, the following issues shall be addressed.

We are glad that the reviewer found the manuscript to be well organized and well written. We are happy that the reviewer found the Supporting information to be relatively in good match with the explanations.

(1) The authors did not make it clear why they utilized three-terminal graphene field effect transistors to implement memristive behavior. What are the advantages of the material and the structure when compared to the traditional oxide-based memristors?

This is a good question posed by the reviewer. The advantages offered by three-terminal memristive devices are not obvious and could stand to be made clearer in the manuscript.

The following discussion on the advantages offered by the three-terminal structure of GFET memristors is included in the revised manuscript: The multiterminal nature of the GFET-based synaptic device allows for it to be modulated by both the V_{DS} programming pulses mentioned in the manuscript and by the V_{BG} applied during read operations. By varying V_{BG} , the resistance of the graphene channel can be modulated, allowing for tuning of the conductance states (weight values) programmed into the device via the V_{DS} pulses as well as the memory ratio between neighboring states. This tuning of weight values through the application of a separate bias could be considered reminiscent of heterosynaptic plasticity in biological neural networks, in which the stimulation of a given neuron causes a change in the strength (weight) of synaptic connections between other, inactivated, neurons [28, 29]. The most well-known example of this mechanism is the existence of modulatory neurons, also known as interneurons. When activated, these neurons

Figure R4: Effects of Back-Gate Voltage on Conductance States in GFETs. The 16 conductance states (memory levels) obtained using write pulses (V_{DS}) with a step size of 0.125 V and duration 1 s are shown for different back-gate voltages: 0 V (blue), +5 V (black), +10 V (grey-blue), +15 V (orange), -5 V (yellow), and -10 V (green). At all non-zero back-gate voltages, the conductance for all 16 states is increased. Additionally, the memory ratio between neighboring states is controllably varied by changing the applied voltage, initially increasing for +/- 5 V and then decreasing at higher voltages as the device saturates. Notably, despite poor conductance switching (memory ratio) in the GFET at higher pulse magnitudes when $V_{BG} = 0$ V, performance is significantly improved for all back-gate voltages tested. It is apparent that applying different back-gate voltages to the GFETs during read and write operations allows for modulation, and even enhancement, of the memory levels achieved by GFETs.

release chemicals known as neuromodulators, differentiated from typical neurotransmitters by their ability to alter synaptic efficacy instead of generating an electrical response. These alterations can last up to several minutes, providing comparatively long-term modulation of synaptic events [30]. In addition, repeated heterosynaptic modulation has, through study, been found to promote the growth/retraction of synaptic connections, creating persistent changes in synaptic weight and contributing to long-term memory formation/storage [31]. This has made the implementation of heterosynaptic plasticity an important goal for developing the next generation of novel neuromorphic systems [29]. The modulation of conductance states afforded by different modulatory bias V_{mod} (V_{BG}) values can be seen in **Supporting Information 5** (included here as Fig. R4). All measurements were conducted on the same GFET using the same V_{DS} pulsing scheme utilized in Figure 2e. Each state was held for 100 s with no observable degradation into neighboring states, indicating good retention for all V_{mod} . The changes in conductance states and memory ratios between adjacent states as a result of changing V_{mod} indicate the ability to implement synaptic potentiation and depression by using the back-gate bias to increase or decrease conductance states

(weight values) independently from the application of programming pulses across the Source and Drain. Thus, the extra degree of freedom offered by the multiterminal design of GFETs allows for synaptic modelling that is not possible in traditional two-terminal synaptic devices, such as those that operate using oxide-based memristors.

(2) The direction of hysteresis and the hysteresis window of some curves in Fig. 1d and 1e are unclear and clear figures should be provided.

These are very helpful observations provided by the reviewer.

Alterations have been made to Fig. 1 to remedy the issue.

(3) The authors claimed that increasing the sweep range appeared to increase the hysteresis window of the GFET until $V_{DSmax} = 5 V$, after which the direction of hysteresis reversed and the hysteresis window began to decrease, which was similar to the situation in opposite voltage polarity. Why the direction of hysteresis reversed? And why did the hysteresis window gradually increase initially and then decrease?

(5) As shown in Fig. 1f and 1g, why higher positive and negative source-drain voltages can induce typical n-type and ambipolar characteristics respectively? The internal mechanism needs to be explained in detail and experimentally verified.

Both of these questions posed by the reviewer are very relevant for establishing/conveying a greater understanding of the underlying switching mechanisms present in GFET memristors.

We have added the following discussion on the hysteresis and conductance switching mechanisms to the revised manuscript: Hysteresis loops of the drain-to-source current have long been noted in graphene and related materials, including graphene oxide and carbon nanotubes (CNTs). This phenomenon has been the subject of numerous studies and is generally attributed to interactions between the materials and trap sites on their substrates and/or extraneous molecules adsorbed on the material surface or at the material/substrate interface [1, 2]. Of these adsorbates, water molecules (H_2O) have seen attention in studies due to their prevalence in most ambient

Figure R2: Hysteresis of PMMA passivated graphene devices. Output characteristics of as-fabricated GFETs following PMMA passivation at a back-gate bias of $V_{BG} = 0$ V for different V_{DS} sweep ranges denoted by V_{DSmax} from a) 1 V to 6.5 V and b) -1 V to -6.5 V in steps of 0.5 V. The arrows denote the sweep direction, with blue representing the forward sweep from 0 V and black representing the backward sweep from V_{DSmax} . Results are similar to those for unpassivated GFETs. As with the output characteristics of unpassivated GFETs shown in Fig. 1d and 1e, hysteresis window initially increases with increasing V_{DSmax} and then reverses direction and starts to decrease. The presence of a passivation layer on the graphene channel establishes that the hysteresis switching, and thus the forming mechanism, does not rely on adsorbates (oxygen molecules, water molecules, etc.) on the graphene free surface. Note that this does not rule out interactions with adsorbates trapped at the graphene/ Al_2O_3 interface. V_{DS} sweeping was repeated following that shown in (a) and (b), with the results being displayed in (c) and (d), respectively. Little-to-no hysteresis was seen, indicating a distinct state change in the GFETs, and confirming the existence of the forming process. Due to the passivation layer, it can be assumed that the mechanisms are thus dominated by interactions at the graphene/ Al_2O_3 interface.

environments, as well as the use of water baths in traditional graphene transfers [3-5]. While surface-bound H_2O can be easily removed via vacuum or the addition of a passivation layer, H_2O trapped at the graphene/substrate interface requires specific treatments to remove and can have significant impact on the electrical properties of the graphene. An investigation by Cho *et al* [6] on the effects of water trapping at the graphene/ Al_2O_3 interface identified two possible adsorption modes for water trapped at the interface: molecular adsorption (in which the oxygen atom is bound to an Al_s site on the substrate surface) and dissociative adsorption (in which the water molecule is split into an OH^- molecule bound to an Al_s site and a H^+ ion bound to an O_s site). The alignment

of H₂O relative to graphene differs between the two modes (parallel for H₂O in molecular adsorption and perpendicular for OH⁻ in dissociative adsorption), leading to differences in the local electrical field, with the field induced by dissociative adsorption being magnitudes larger than that induced by molecular adsorption. The stronger dissociative field in turn leads to a higher planar-averaged charge density and p-type doping of the graphene [2, 5, 6].

Based on the distinctly p-type nature of the GFETs tested and discussed in this manuscript, it is reasonable to assume that it is a result of dissociative adsorption of H₂O trapped at the graphene/Al₂O₃, most likely as a result of the graphene transfer process discussed in the *Methods* section. Similar processes have been noted to result in trapped water adlayers at the interfaces of graphene and a number of different substrates [3, 6, 7]. Stemming from this, it is also reasonable to assume that the hysteresis shown in Fig. 1d and 1e is primarily caused by the trapped H₂O as well. To explore this phenomenon further, effort was made to observe the effects of passivation upon the demonstrated GFET hysteresis switching, a separate set of GFETs was fabricated on a separate Al₂O₃ substrate and passivated *via* the deposition of 120 nm of PMMA. Following passivation, the hysteresis switching tests discussed and demonstrated in Fig. 1d and 1e were performed upon the passivated devices. The results for these tests are demonstrated in ***Supporting Information 2*** (included here as Fig. R2). The hysteresis seen for the positive and negative sweeps in Fig. R2a and R2b closely resembles that seen in Fig. 1d and 1e, respectively. This indicates that the hysteresis and hysteresis switching is not tied to any adsorbates on the free surface of the graphene channel.

However, this does not rule out contributions from adsorbates trapped at the graphene/Al₂O₃ interface. Previous studies, such as that by Woong Kim *et al* [35], have established that hysteresis due to adsorption of water at the interface can persist following surface passivation. Based on these observations, the forming process discussed in this paper is believed to be a result of switching between different adsorption modes for water molecules trapped at the graphene/Al₂O₃ interface. Following fabrication, these molecules are believed to be dissociatively adsorbed on account of the distinctly p-type nature of the transfer characteristics for all GFETs tested as well as the noticeable hysteresis when observing the swept output characteristics. The increase in the drain bias applied during these sweeps is believed to induce a transition to molecular adsorption of the

water molecules. The OH^- molecule and H^+ ion bound to an Al_S site and O_S site, respectively, on the Al_2O_3 surface would recombine and bind to an Al_S site, with the OH-bonds of the resulting H_2O molecule lying relatively parallel to the plane of the graphene. This is supported by the transition of the GFET transfer characteristics following each sweep; as V_{DSmax} increases in magnitude, V_{Dirac} shifts more and more negative, causing the transfer characteristics to become either ambipolar (for negative bias pulsing) or n-type (for positive bias pulsing). While the GFET is then able to demonstrate analog switching between the n-type and ambipolar states, it is unable to return to the original p-type characteristics indicative of dissociative adsorption. In addition, following the initial bias sweeping, any subsequent sweeping fails to demonstrate any significant hysteresis, a known characteristic of molecular adsorption. This can be seen in ***Supporting Information 2c*** and ***2d***.

The conductance switching demonstrated in GFETs following the forming process, as highlighted by Fig. 1d through 1g, is believed to be the result of dipole moment switching due to the generated electric field. Such effects have been shown to result in threshold and conductance shifting in field effect transistors with interfacial dipole monolayers, leading to the development of distinct memory states [9-13]. Following the forming process (the transition from dissociative adsorption of water molecules at the interface to molecular adsorption), the water molecules are randomly oriented due to the uncoordinated nature of Al_S states at the Al_2O_3 surface [6]. In this state, the local electric field generated by the water molecules is far weaker than in dissociative adsorption, owing to the interference caused by the random orientation of neighboring dipoles. As a result, the graphene tends to display ambipolar transfer characteristics as opposed to its initial p-type characteristics. Previous studies have shown that interfacial water molecules at a graphene surface can be reoriented through the application of an external electric field [14, 15]. This polarizes the water molecules and can align their dipoles due to their preference for an orientation parallel to the electric field, enhancing the local electric field and increasing conductance of the graphene channel [16]. Experimentally, this phenomenon is reflected by the increase in conductance observed when positive bias pulses are applied to the GFETs through the drain, as demonstrated in Fig. 2g through 2j. When negative bias pulses are applied, the water molecules are oppositely polarized leading to reorientation. This is reflected by the decrease in conductance through negative bias pulsing shown in Fig. 2b through 2e.

(4) Are there any defects at the interface between alumina and graphene? If the defects exist, will they affect the channel conductance when the gate voltage is applied?

We have added the following discussion on interface defects to the revised supplemental materials as *Supporting Information 10*: As stated in the manuscript, the dominant interaction at the graphene/ Al_2O_3 interface is believed to be that involving trapped water molecules. When the molecules are dissociatively adsorbed to the Al_2O_3 surface, the strong negative electric field generated has significant effect of the band structure of the graphene channel, effectively raising the valence band and inducing p-type doping. It has also been shown to induce interbanding between the graphene and O_s bands at the Dirac point. When the water molecules are shifted to a molecularly adsorbed state, the local electric field is removed, lowering the valence band and returning the Dirac point of the graphene to the Fermi level [6]. Experimentally, this is reflected by the transition of the GFETs to an ambipolar state from their initial p-type characteristics. While adsorbed water molecules at the interface are believed to be the dominant mechanism in the shown hysteresis and conductance switching, this does not rule out the existence of Al_2O_3 defects and trap states or other adsorbates, such as PMMA residue from the transfer and lithography processes, that may also affect the graphene channel. However, based on past studies of the effects of PMMA residue, such as heavy p-type doping, low conductivity due to carrier scattering, and large hysteresis from the introduction of trap states, PMMA residue should not be a major concern [32]. The rather moderate p-type doping demonstrated by the GFETs, as well as the fact that both it and the demonstrated hysteresis are removable by applying bias pulses through the drain, indicate that resist residues have very little effect, if any, on the graphene characteristics. While the argument may be made these phenomena may be in part due to current-induced cleaning of the graphene channel [33], GFETs have been shown to return to their original p-type characteristics if left unused for long (on the order of weeks to months) periods of time and still remain programmable, ruling out PMMA residue as the dominant interaction. In addition, most trap states at interfaces with Al_2O_3 are attributed to dangling Al-O bonds at the oxide surface [34]. As most of these states would be occupied by the adsorbed water molecules, the overall charge trapping should remain small.

(6) *The operating voltage of 4.5 V is still relatively large for memristive devices and a switching endurance more than 200 cycles is also insufficient for neuromorphic computing. How to further decrease the voltage and improve the endurance ability?*

We have addressed the operating voltage of GFETs and its scalability as part of the following discussion on GFET scalability in the revised manuscript: To investigate the scalability of GFET memristors, several sets of GFETs featuring reduced channel lengths (L), ranging from 200 nm to 800 nm in steps of 200 nm, were fabricated on a separate Al₂O₃ substrate using the same fabrication processes discussed in the *Methods* section. Half of each set was fabricated with a channel width (W) of 1 μm, matching that of the originally tested devices, while the other half was fabricated with a channel width matching the channel length. The device layout that covered the smallest area while remaining functional was found to be that with L = 400 nm and W = 1 μm, for a total area of 0.4 μm². In addition to the reduced area, the devices demonstrated the ability to shift conductance states at lower pulse magnitudes than the 1 μm² devices detailed previously, indicating a channel length/area dependency for the conductance switching mechanism in GFETs.

Supporting Information 6 (included here as Fig. R3) demonstrates the shifting from initial p-type characteristics indicative of the forming process at lower voltages than was necessary for the 1 μm² devices. This can be explained as an effect of the shorter channel length increasing the electric field generated by each pulse, allowing for reorientation of the water molecule dipoles trapped at the graphene/Al₂O₃ interface at lower biases. The shift from n-type to ambipolar characteristics when negative low voltage pulses are applied, as shown in ***Supporting Information 6b***, supports this theory. Interestingly, the 400 nm channel length devices continued to display high endurance even at high pulse magnitudes. ***Supporting Information 6c*** and ***6d*** display the shifting of the Dirac point when positive and negative pulses of magnitude 4 V and 5 V, respectively, are applied. Insets show the endurance at each pulse magnitude for 200 cycles. The 5 V pulse cycling, which was noted as the maximum sustainable voltage for the 1 μm channel length devices, shows similar memory ratio and endurance to that of the 1 μm channel length devices discussed earlier. These results indicate that a wider range of pulse magnitudes can be used for shorter channel devices, potentially increasing the number of distinct achievable memory states while retaining similar endurance. In contrast, all 200 nm channel length devices tested were found to be intrinsically

Figure R3: Forming, Conductance Switching, and Endurance of Scaled GFET. a) Transfer characteristics of a p-type GFET with channel length 400 nm and channel width 1 μm when positive write voltage pulses of duration 1 s with increasing magnitude are applied. Starting at +1 V and increasing to +3.4 V in steps of +0.2 V. The transition from p-type to n-type characteristics shown is similar to that demonstrated by the forming process displayed in Fig. 1, indicating the presence of memristive switching in scaled GFETs. The lower voltages needed to achieve forming for the scaled GFET indicates strong channel length/area dependence of the forming mechanism, implying that it is electric field dominated. b) Transfer characteristics of the same GFET when negative write pulses of the same magnitudes and step sizes as (a) are applied. The noticeable shift to more ambipolar characteristics from the n-type characteristics programmed in (a) confirms memristive switching in scaled GFETs. c-d) Transfer characteristics of the GFET for +/- c) 4 V and d) 5 V switching. Despite shifting beginning at lower voltages, scaled GFET demonstrates the ability to be programmed up to the maximum voltages established for longer channel devices, indicating that shorter channel devices may be able to utilize a wider distribution of conductance states (memory levels). Insets display endurance testing over 200 cycles for respective pulse magnitudes with no observable degradation. V_{Dirac} shifting and endurance results of (c) and (d) are comparable to those of longer channel devices, indicating that GFETs are able to be scaled without adverse effects on device characteristics.

ambipolar and displayed little-to-no shifting when programming pulses were applied. These results were taken to support our initial hypothesis regarding the role of water molecule adsorption and dipole alignment at the interface regarding changing conductance states. It is believed that the small area of the 200 nm channel length devices did not allow for sufficient water molecules to be trapped to demonstrate p-doping via dissociative adsorption or conductance shifting via dipole

realignment. The channel scalability demonstrated also shows promise for large-scale integration of GFETs into crossbar-array architectures. As shown and discussed in **Supporting Information 7**, the nature of the programming phenomenon of GFETs, bias pulsing through the drain, allows for electrostatic isolation of devices in close proximity to one another despite the presence of a global back-gate. Together with the aforementioned scaling, this indicates the potential for high integration density of GFET memristors, offering an attractive alternative for close-packed memristive device architectures such as dense crossbar-arrays.

We have added the following information on long-term GFET capabilities to the revised supplemental material as *Supporting Information 4*: Long-term capabilities of GFETs. Fig. R5a

shows the memory ratio and retention measured for a total duration of 1000 s for GFET programmed into 8 memory levels using negative write pulses of step size 0.5 V and duration 1 s. Memory levels remain distinct

for the entire read process, demonstrating good retention on par with a number of other published works concerning 2D material synaptic devices [18, 25, 27, 35]. Fig. R5n shows the conductance states

over 500 cycles of SET and RESET pulses of magnitude 5 V. Memory ratio remains consistent between the high and low conductance states over the entirety of the cycling

process, signifying long term endurance. Endurance can be further improved by using lower pulse magnitudes during switching. As demonstrated in **Supporting Information 5**, shorter channel length devices require smaller biases to induce conductance shifting, pointing towards higher endurance while still retaining reasonable memory ratios between the maximum and minimum conductance states.

Figure R5. Long-term capabilities of GFETs. a) Memory ratio and retention measured for a total duration of 1000 s for GFET programmed into 8 memory levels using negative write pulses of step size 0.5 V and duration 1 s. Memory levels remain distinct for the entire read process, demonstrating good retention on par with a number of other published works concerning 2D material synaptic devices [18, 25, 27, 35]. b) Conductance states over 500 cycles of SET and RESET pulses of magnitude 5 V. Memory ratio remains consistent between the high and low conductance states over the entirety of the cycling process, signifying long term endurance.

(7) What's the energy consumption of graphene field effect transistors?

We have added the following comment on the power/energy consumption of GFET memristors to the revised manuscript: Power consumption for the GFETs is approximately 5 mW for write operations at a pulse magnitude of 5 V and less than 40 nW for read operations at a read voltage of 10 mV. Using of pulse time of 1 s, this establishes a switching energy of approximately 5 mJ.

(8) There seems to be no obvious error difference in Fig. 4e and 4f when N increase from 2 to 4. Please make more comments.

The reviewer is correct to observe that the overall error difference between Fig. 4e and 4f is minimal. This is an artifact from an earlier draft on the manuscript that was overlooked during editing and we sincerely apologize for its inclusion.

We have added the following discussion addressing the percent errors for the device programming demonstrated in Figure 4 to the revised manuscript: Despite the increase in the value of N being utilized, the overall error difference between Fig. 4e and 4f is minimal. This is due to the relative position of the conductance values being targeted, $G_1 = 215 \mu\text{S}$ and $G_2 = 155 \mu\text{S}$. The allowed conductance states for the two GFETs being utilized when they are restricted to two memory levels ($N = 2$) are $230 \mu\text{S}$ and $140 \mu\text{S}$. Each achievable state differs from its targeted value by $15 \mu\text{S}$, resulting in the error shown in Fig. 4e when G_1 is set to $230 \mu\text{S}$ and G_2 to $140 \mu\text{S}$. When the GFETs are allowed to operate using four memory levels ($N = 4$), G_1 and G_2 are reprogrammed in an effort to reduce the error between the targeted and actual conductance values, with G_1 being set to $200 \mu\text{S}$ and G_2 to $170 \mu\text{S}$. However, these states, like those when $N = 2$, are $15 \mu\text{S}$ off from the targeted conductance values. As demonstrated by Fig. 4f, this means that while the error distribution is shifted slightly due to the relative position of the conductance states changing, the overall accuracy of the synapse is not improved. This serves to highlight a drawback of uniform weight distribution and its implementation using devices with discrete memory states, such as oxide-based memristors. When the desired weight (conductance) value lies between set states, it can be very difficult for the system to reach it unless it utilizes a very large number of

memory states. The analog nature of GFET memristors, on the other hand, allows for precise programming of any weight (conductance) value within the distribution of conductance states. The experimentally obtained output current for when the GFETs are directly programmed to the nearest achievable conductance states, $G_1^K = 214 \mu\text{S}$ and $G_2^K = 156 \mu\text{S}$, can be seen in Fig. 4g. The error between this output current and the expected output current shown in Fig. 4b is displayed by Fig. 4h. As would be expected, the error is significantly reduced.

(9) There are some format mistakes that need to be dealt with in the citation.

We thank the reviewer for this observation and have edited the citations accordingly.

Rebuttal References

- [1] Z.-M. Liao, B.-H. Han, Y.-B. Zhou, and D.-P. Yu, "Hysteresis reversion in graphene field-effect transistors," *The Journal of chemical physics*, vol. 133, p. 044703, 2010.
- [2] H. Wang, Y. Wu, C. Cong, J. Shang, and T. Yu, "Hysteresis of electronic transport in graphene transistors," *ACS nano*, vol. 4, pp. 7221-7228, 2010.
- [3] T. O. Wehling, A. I. Lichtenstein, and M. I. Katsnelson, "First-principles studies of water adsorption on graphene: The role of the substrate," *Applied Physics Letters*, vol. 93, p. 202110, 2008.
- [4] G. Hong, Y. Han, T. M. Schutzius, Y. Wang, Y. Pan, M. Hu, *et al.*, "On the mechanism of hydrophilicity of graphene," *Nano letters*, vol. 16, pp. 4447-4453, 2016.
- [5] F. Yavari, C. Kritzinger, C. Gaire, L. Song, H. Gulapalli, T. Borca-Tasciuc, *et al.*, "Tunable bandgap in graphene by the controlled adsorption of water molecules," *small*, vol. 6, pp. 2535-2538, 2010.
- [6] S. B. Cho, S. Lee, and Y.-C. Chung, "Water trapping at the graphene/Al₂O₃ interface," *Japanese Journal of Applied Physics*, vol. 52, p. 06GD09, 2013.
- [7] C. Goldmann, D. Gundlach, and B. Batlogg, "Evidence of water-related discrete trap state formation in pentacene single-crystal field-effect transistors," *Applied Physics Letters*, vol. 88, p. 063501, 2006.
- [8] W. Kim, A. Javey, O. Vermesh, Q. Wang, Y. Li, and H. Dai, "Hysteresis caused by water molecules in carbon nanotube field-effect transistors," *Nano Letters*, vol. 3, pp. 193-198, 2003.
- [9] B. Traoré, P. Blaise, E. Vianello, H. Grampeix, S. Jeannot, L. Perniola, *et al.*, "On the origin of low-resistance state retention failure in HfO₂-based RRAM and impact of doping/alloying," *IEEE Transactions on Electron Devices*, vol. 62, pp. 4029-4036, 2015.
- [10] W. Ou-Yang, X. Chen, M. Weis, T. Manaka, and M. Iwamoto, "Tuning of threshold voltage in organic field-effect transistor by dipole monolayer," *Japanese Journal of Applied Physics*, vol. 49, p. 04DK04, 2010.
- [11] C. Celle, C. Suspène, M. Ternisien, S. Lenfant, D. Guérin, K. Smaali, *et al.*, "Interface dipole: Effects on threshold voltage and mobility for both amorphous and poly-crystalline organic field effect transistors," *Organic Electronics*, vol. 15, pp. 729-737, 2014.
- [12] K. Pernstich, S. Haas, D. Oberhoff, C. Goldmann, D. Gundlach, B. Batlogg, *et al.*, "Threshold voltage shift in organic field effect transistors by dipole monolayers on the gate insulator," *Journal of Applied Physics*, vol. 96, pp. 6431-6438, 2004.
- [13] S. Vasudevan, N. Kapur, T. He, M. Neurock, J. M. Tour, and A. W. Ghosh, "Controlling transistor threshold voltages using molecular dipoles," *Journal of Applied Physics*, vol. 105, p. 093703, 2009.
- [14] H. Ren, L. Zhang, X. Li, Y. Li, W. Wu, and H. Li, "Interfacial structure and wetting properties of water droplets on graphene under a static electric field," *Physical Chemistry Chemical Physics*, vol. 17, pp. 23460-23467, 2015.
- [15] C. Melios, C. E. Giusca, V. Panchal, and O. Kazakova, "Water on graphene: review of recent progress," *2D Materials*, vol. 5, p. 022001, 2018.
- [16] M. S. Fernández, F. Peeters, and M. Neek-Amal, "Electric-field-induced structural changes in water confined between two graphene layers," *Physical Review B*, vol. 94, p. 045436, 2016.
- [17] M. S. Choi, G. H. Lee, Y. J. Yu, D. Y. Lee, S. H. Lee, P. Kim, *et al.*, "Controlled charge trapping by molybdenum disulphide and graphene in ultrathin heterostructured memory devices," *Nat Commun*, vol. 4, p. 1624, 2013.
- [18] H. Tian, L. Zhao, X. Wang, Y. W. Yeh, N. Yao, B. P. Rand, *et al.*, "Extremely Low Operating Current Resistive Memory Based on Exfoliated 2D Perovskite Single Crystals for Neuromorphic Computing," *ACS Nano*, vol. 11, pp. 12247-12256, Dec 26 2017.

- [19] Y. Shi, C. Pan, V. Chen, N. Raghavan, K. Pey, F. Puglisi, *et al.*, "Coexistence of volatile and non-volatile resistive switching in 2D h-BN based electronic synapses," in *2017 IEEE International Electron Devices Meeting (IEDM)*, 2017, pp. 5.4. 1-5.4. 4.
- [20] M. T. Sharbati, Y. Du, J. Torres, N. D. Ardolino, M. Yun, and F. Xiong, "Low-Power, Electrochemically Tunable Graphene Synapses for Neuromorphic Computing," *Adv Mater*, p. e1802353, Jul 23 2018.
- [21] V. K. Sangwan, H. S. Lee, H. Bergeron, I. Balla, M. E. Beck, K. S. Chen, *et al.*, "Multi-terminal memtransistors from polycrystalline monolayer molybdenum disulfide," *Nature*, vol. 554, pp. 500-504, Feb 21 2018.
- [22] J. Zhu, Y. Yang, R. Jia, Z. Liang, W. Zhu, Z. U. Rehman, *et al.*, "Ion Gated Synaptic Transistors Based on 2D van der Waals Crystals with Tunable Diffusive Dynamics," *Adv Mater*, vol. 30, p. e1800195, May 2018.
- [23] D. Li, B. Wu, X. Zhu, J. Wang, B. Ryu, W. D. Lu, *et al.*, "MoS₂ Memristors Exhibiting Variable Switching Characteristics toward Biorealistic Synaptic Emulation," *ACS Nano*, vol. 12, pp. 9240-9252, Sep 25 2018.
- [24] Y. Shi, X. Liang, B. Yuan, V. Chen, H. Li, F. Hui, *et al.*, "Electronic synapses made of layered two-dimensional materials," *Nature Electronics*, vol. 1, pp. 458-465, 2018.
- [25] W. Huh, S. Jang, J. Y. Lee, D. Lee, D. Lee, J. M. Lee, *et al.*, "Synaptic Barristor Based on Phase-Engineered 2D Heterostructures," *Adv Mater*, vol. 30, p. e1801447, Aug 2018.
- [26] X. Zhu, D. Li, X. Liang, and W. D. Lu, "Ionic modulation and ionic coupling effects in MoS₂ devices for neuromorphic computing," *Nat Mater*, vol. 18, pp. 141-148, Feb 2019.
- [27] C. He, J. Tang, D. S. Shang, J. Tang, Y. Xi, S. Wang, *et al.*, "Artificial Synapse Based on van der Waals Heterostructures with Tunable Synaptic Functions for Neuromorphic Computing," *ACS Appl Mater Interfaces*, vol. 12, pp. 11945-11954, Mar 11 2020.
- [28] W. Huh, S. Jang, J. Y. Lee, D. Lee, D. Lee, J. M. Lee, *et al.*, "Synaptic Barristor Based on Phase-Engineered 2D Heterostructures," *Adv Mater*, vol. 30, p. e1801447, Aug 2018.
- [29] C. He, J. Tang, D. S. Shang, J. Tang, Y. Xi, S. Wang, *et al.*, "Artificial Synapse Based on van der Waals Heterostructures with Tunable Synaptic Functions for Neuromorphic Computing," *ACS Appl Mater Interfaces*, vol. 12, pp. 11945-11954, Mar 11 2020.
- [30] G. S. Lynch, T. Dunwiddie, and V. Gribkoff, "Heterosynaptic depression: a postsynaptic correlate of long-term potentiation," *Nature*, vol. 266, pp. 737-739, 1977.
- [31] C. H. Bailey, M. Giustetto, Y.-Y. Huang, R. D. Hawkins, and E. R. Kandel, "Is heterosynaptic modulation essential for stabilizing hebbian plasticity and memory," *Nature Reviews Neuroscience*, vol. 1, pp. 11-20, 2000.
- [32] Y. Dan, Y. Lu, N. J. Kybert, Z. Luo, and A. C. Johnson, "Intrinsic response of graphene vapor sensors," *Nano letters*, vol. 9, pp. 1472-1475, 2009.
- [33] J. Moser, A. Barreiro, and A. Bachtold, "Current-induced cleaning of graphene," *Applied Physics Letters*, vol. 91, p. 163513, 2007.
- [34] M. M. Furchi, D. K. Polyushkin, A. Pospischil, and T. Mueller, "Mechanisms of photoconductivity in atomically thin MoS₂," *Nano letters*, vol. 14, pp. 6165-6170, 2014.
- [35] Z. Vasicek, V. Mrazek, and L. Sekanina, "Automated Circuit Approximation Method Driven by Data Distribution," in *2019 Design, Automation & Test in Europe Conference & Exhibition (DATE)*, 2019, pp. 96-101.

REVIEWER COMMENTS

Reviewer #3 (Remarks to the Author):

The authors have addressed all the issues. However, in my opinion, there are still two questions need to be revised before accepted.

1, According to figure 1d and e, the relatively large hysteresis still exists at $V_{DS}=6.5V$ after the forming process at 5.5 V. It seems contradictory to the results in Figure R2c and d with almost no hysteresis. Based on the forming mechanism about H₂O adsorption, dissociative adsorption can be transited to molecular adsorption during the forming process. Therefore, the hysteresis at 6.5V may be contradictory to the known characteristic of molecular adsorption.

2, What is the value of the gate voltage used to read I_{ds} in Figure 2b to j ? According to Figure R4, the linearity of the I_{ds} can be improved if the optimized gate voltage was applied. So, whether the accuracy in Figure 4 will be raised furtherly under an optimized gate voltage?

Response to Reviewers

Reviewer 3:

The authors have addressed all the issues. However, in my opinion, there are still two questions need to be revised before accepted.

1, According to figure 1d and e, the relatively large hysteresis still exists at $V_{DS}=6.5V$ after the forming process at 5.5 V. It seems contradictory to the results in Figure R2c and d with almost no hysteresis. Based on the forming mechanism about H_2O adsorption, dissociative adsorption can be transitioned to molecular adsorption during the forming process. Therefore, the hysteresis at 6.5V may be contradictory to the known characteristic of molecular adsorption.

We are glad that the reviewer found the revisions satisfactory. The reviewer has raised an excellent point related to the work. The presence of hysteresis following the completion of the forming process in Fig. 1 may seem contradictory to the results shown in Fig. S2. However, it is our belief that the hysteresis shown in Fig. 1 and Fig. S2 following the forming process is the result of remanent polarization, the dipole polarization remaining after the reorienting electric field is removed.

We have added the following brief discussion on hysteresis to the revised supplemental material as *Supporting Information 11*: It is important to note the significant hysteresis at high V_{DSmax} demonstrated in Fig. 1d and 1e and Fig. S2a and S2b. Although the forming process was demonstrated to occur at lower V_{DSmax} values, as denoted by the switching of the sweep direction, the hysteresis remains high following forming. Based on the postulated forming mechanism discussed in the manuscript (i.e. the switching of water molecules at the graphene/ Al_2O_3 interface from dissociative adsorption to molecular adsorption), the post-forming hysteresis should be negligible, akin to that seen in Fig. S2c and S2d. To investigate the origins of the hysteresis at high V_{DSmax} , the GFETs used for the tests shown in Fig. S2 were subjected to large (6.5 V) programming pulses of negative and positive polarity immediately following the tests shown in Fig. S2c and S2d, respectively. The output characteristics of the GFETs were then measured for different V_{DS} sweep ranges, as in Fig. 1 and Fig. S2, with the results being shown in Fig. S11. The results show hysteresis greater than that displayed in Fig. S2c and S2d, though less than that displayed in Fig.

Figure S11: Hysteresis of reset graphene devices. Output characteristics of GFETs at a back-gate bias of $V_{BG} = 0$ V for different V_{DS} sweep ranges denoted by V_{DSmax} from a) 1 V to 6.5 V and b) -1 V to -6.5 V in steps of 0.5 V. Measurements were taken following those shown in Fig. S2c and S2d, corresponding to (a) and (b), respectively, with the GFETs first being reset using a programming pulse of opposite polarity, magnitude 6.5 V, and duration 1 s.

1 and Fig. S2a and S2b. This, in addition to the noticeable lack of sweep direction switching, indicates the origin of the hysteresis is distinct from the forming mechanism seen in GFETs. Based on the memristive mechanism discussed in the manuscript (i.e. the reorientation of water dipoles at the graphene/ Al_2O_3 interface), the hysteresis is believed to be a result of differences between the polarization of dipoles following each V_{DSmax} and the remanent polarization of the GFET (i.e. the polarization of the dipoles following the removal of the previous polarizing electric field). Due to initial disorder in the orientation of dipoles following the forming process [1], the hysteresis in the initial V_{DS} sweep testing, as shown in Fig. 1 and Fig. S2a and S2b, is relatively large. However, when the tests are redone without resetting the

GFETs, as in Fig. S2c and S2d, the dipoles are already polarized, resulting in no hysteresis. When the devices are reset using programming pulses of opposite polarity (Fig. S11), the repolarization of the dipoles during following V_{DS} sweeps causes the hysteresis to return, albeit at a smaller magnitude than before the forming process, as would be expected of molecularly adsorbed water molecules when compared to dissociatively adsorbed molecules.

2, What is the value of the gate voltage used to read I_{ds} in Figure 2b to j ? According to Figure R4, the linearity of the I_{ds} can be improved if the optimized gate voltage was applied. So, whether the accuracy in Figure 4 will be raised furtherly under an optimized gate voltage?

We thank the reviewer for pointing out that the read gate voltage for the results shown in Fig. 2 was never specified. The manuscript has been modified to accurately reflect the read gate voltage, $V_{BG} = 0$ V. The results shown in Fig. 4 were also obtained at a read gate voltage of 0 V. Based on the results shown in **Supporting Information 5**, varying the back-gate voltage has such effects on GFET operation as changing the range of achievable conductance values and increasing/decreasing the memory ratio between neighboring conductance states. While proper optimization of these factors could result in increased VMM accuracy for devices utilizing discrete memory states, such as oxide-based memristors, that is not believed to be the case for the GFET memristive devices discussed in the manuscript. In the discussion pertaining to the results displayed in Fig. 4, it is established that the accuracy shown is enabled by the analog nature of GFETs, which allows for direct programming of a device to a desired conductance state. As a result, changing the memory ratio between states would have no effect on the VMM accuracy as the desired state would still be programmed directly. However, the effects of varying V_{BG} may still be utilized for optimization of VMM operations. The ability to change the achievable range of conductance values of a GFET may be used to allow the GFET to reach weight values outside of its intrinsic range, allowing greater differentiation between synapses. Additionally, varying V_{BG} can change the conductance value of the set state, allowing for tuning of weight values without requiring the use of programming pulses. Proper utilization of this could allow for less SET/RESET programming cycles and thus longer device lifetimes. Finally, while changing the memory ratio between neighboring states will not affect programming accuracy, it would change the pulse magnitudes needed to program different states. Proper optimization could thus be used to reduce power/energy consumption of GFET write steps by minimizing the necessary pulse magnitudes.

References

- [1] Cho SB, Lee S, Chung Y-C. Water Trapping at the Graphene/ Al_2O_3 Interface. *Japanese Journal of Applied Physics* **52**, (2013).

REVIEWERS' COMMENTS

Reviewer #3 (Remarks to the Author):

The authors have addressed all the issues. Now this paper can be accepted by Nature Communications.